# HDAC3 restrains CD8-lineage genes to maintain a bi-potential state in CD4+CD8+ thymocytes for CD4-lineage commitment

Rachael Laura Philips[1], Jeong-Heon Lee[2], Krutika Gaonkar[3], Pritha Chanana[3], Ji Young Chung[1], Sinibaldo R Romero Arocha[1], Aaron Schwab[1], Tamas Ordog[2], Virginia Smith Shapiro[1]*

[1]Department of Immunology, Mayo Clinic, Rochester, United States; [2]Epigenomics Program, Center for Individualized Medicine, Mayo Clinic, Rochester, United States; [3]Department of Health Science Research, Division of Biostatistics and Informatics, Mayo Clinic, Rochester, United States

**Abstract** CD4 and CD8 T cells are vital components of the immune system. We found that histone deacetylase 3 (HDAC3) is critical for the development of CD4 T cells, as HDAC3-deficient DP thymocytes generate only CD8SP thymocytes in mice. In the absence of HDAC3, MHC Class II-restricted OT-II thymocytes are redirected to the CD8 cytotoxic lineage, which occurs with accelerated kinetics. Analysis of histone acetylation and RNA-seq reveals that HDAC3-deficient DP thymocytes are biased towards the CD8 lineage prior to positive selection. Commitment to the CD4 or CD8 lineage is determined by whether persistent TCR signaling or cytokine signaling predominates, respectively. Despite elevated IL-21R/γc/STAT5 signaling in HDAC3-deficient DP thymocytes, blocking IL-21R does not restore CD4 lineage commitment. Instead, HDAC3 binds directly to CD8-lineage promoting genes. Thus, HDAC3 is required to restrain CD8-lineage genes in DP thymocytes for the generation of CD4 T cells.
DOI: https://doi.org/10.7554/eLife.43821.001

*For correspondence:
shapiro.virginia1@mayo.edu

**Competing interests:** The authors declare that no competing interests exist.

## Introduction

T cells are essential components of the adaptive immune system that directly kill pathogen-infected cells (CD8 cytotoxic T cells) or orchestrate other immune cells in host defense (CD4 helper T cells). The decision to become either a CD8 or CD4 T cell occurs in the thymus. Proper lineage choice involves the coordination of various cellular signals with the regulation of gene expression by lineage dependent transcription factors. According to the kinetic signaling model (reviewed in *Singer et al., 2008*), lineage choice is determined by TCR signaling persistence versus cytokine signaling. After positive selection, thymocytes downregulate the CD8 coreceptor to become 'intermediate' CD4+-CD8lo thymocytes to test for coreceptor specificity. MHC class II-restricted thymocytes experience persistent TCR signaling, upregulate the CD4-lineage determining transcription factor ThPOK, and commit to the CD4 lineage. Alternatively, TCR signaling ceases with MHC class I-restricted thymocytes, allowing for common gamma chain cytokine signaling (mainly IL-7R) to predominate, induction of the CD8-lineage determining transcription factor Runx3, and commitment to the CD8-lineage.

While many transcription factors have been identified in regulate lineage choice, less is known about the contribution of different chromatin modifiers. HDAC3 is a critical regulator of gene expression. HDAC3 exists in a large (~270 kDa) multiprotein co-repressor complex containing the nuclear receptor corepressor (NCOR) and/or its homolog silencing mediator of retinoic and thyroid receptors (SMRT) (reviewed by *Karagianni and Wong, 2007*). HDAC3 can also associate with p300 coactivator complexes (*Zhang et al., 2016*) to modulate histone acetyltransferase activity. HDAC3

does not have a DNA-binding domain, therefore it must be recruited to different areas of the genome via interactions with different transcription factors. Within the nucleus, HDAC3 can be found at promoters or enhancers and functions to deacetylate specific lysine residues on histones H3 and H4. Therefore, HDAC3 is a dynamic protein that represses gene expression depending upon the complexes HDAC3 interacts with and where it is recruited in the genome.

HDAC3 is ubiquitously expressed and plays important and specific roles in cell biology and disease. Genomic deletion of HDAC3 is embryonically lethal due to defects in gastrulation (*Montgomery et al., 2008*), therefore conditional knockout models are necessary to elucidate HDAC3 function in particular cell types. Within the immune system, HDAC3 is important for the maintenance of hematopoietic stem cells (*Summers et al., 2013*), peripheral T cell maturation (*Hsu et al., 2015*), regulatory T cell development and function (*Wang et al., 2015*), iNKT cell development and differentiation (*Thapa et al., 2017*), B cell development (*Stengel et al., 2017*), and macrophage function (*Mullican et al., 2011*).

In addition, HDAC3 is required during thymocyte development for positive selection and CD4-lineage development (*Philips et al., 2016*; *Stengel et al., 2015*). Mice with CD2-icre-mediated deletion of *loxP*-flanked *Hdac3* (CD2-icre HDAC3-cKO, referred to as HDAC3-cKO) exhibit a positive selection block due to the failure to down-regulate RORγt (*Philips et al., 2016*). Deletion of RORγt and transgenic expression of Bcl-xl corrects the positive selection defect in HDAC3-cKO mice (RORγt-KO Bcl-xl Tg HDAC3-cKO, referred to as RB3), as RORγt-KO corrects for the failure to down-regulate RORγt and the Bcl-xl transgene restores the DP-survival defect in RORγt-KO mice (*Philips et al., 2016*; *Sun et al., 2000*). However, thymocytes almost exclusively develop into CD8SP with very few CD4SP (*Philips et al., 2016*). RORγt-KO Bcl-xl Tg mice exhibited normal numbers of CD4SP and CD8SP thymocytes (*Philips et al., 2016*), demonstrating that the defect in CD4-lineage development in RB3 mice is specific to HDAC3.

Here, we elucidate how HDAC3 is essential for CD4-lineage commitment. The failure to generate CD4-lineage cells is due to the redirection of MHC class II-restricted thymocytes to the CD8-lineage. After positive selection, HDAC3-deficient thymocytes exhibit a failure to induce the CD4-lineage program and acceleration of commitment towards the CD8-lineage. HDAC3 binds to regulatory elements of CD8-lineage promoting genes *Runx3* and *Patz1* in DP thymocytes, and deletion of HDAC3 results in an increase in histone acetylation and mRNA levels. In addition, HDAC3 binds to *Runx3* and *Patz1* regions in OT-II CD4SP thymocytes and was absent in OT-I CD8SP thymocytes. Therefore, our data demonstrates that HDAC3 functions in DP thymocytes to repress CD8-lineage genes in order for DP thymocytes to maintain a bi-potential state.

## Results

### MHC class II restricted thymocytes are redirected to the CD8-lineage when HDAC3 is absent

To examine why few CD4SP thymocytes develop in RB3 mice, T cell development was examined in RB3 mice expressing a transgene encoding the OT-II MHC class II-restricted TCR (*Barnden et al., 1998*). While mature SP thymocytes and splenic T cells from OT-II mice were primarily CD4$^+$ cells, OT-II RB3 mice generated CD8$^+$ cells (*Figure 1A*). In addition, mature SP thymocytes from OT-II RB3 mice expressed Runx3 and lacked ThPOK expression (*Figure 1B*), demonstrating that MHC class II-restricted thymocytes are re-directed to the CD8-lineage in OT-II RB3 mice. Control OT-II RORγt-KO Bcl-xl tg (OT-II RB) mice showed normal commitment to the CD4 lineage (*Figure 1A–B*), indicating that the redirection observed in OT-II RB3 mice is a result of HDAC3 deletion. Redirection was not dependent on Bcl-xl expression, as OT-II RORγt-KO HDAC3-cKO mice also showed redirection to the CD8-lineage compared to OT-II RORγt-KO mice (*Figure 1C*), despite a reduction in the number of SP thymocytes in RORγt-deficient mice. To determine whether redirection also led to the acquisition of CD8-lineage function, SP thymocytes from OT-II RB3 mice were TCR stimulated and analyzed for Granzyme b and Perforin production. MHC class I-restricted OT-I cells, as well as OT-II cells and OT-II RB cells, were used as positive and negative controls, respectively. As expected, OT-I cells produced Granzyme b and Perforin, while OT-II and OT-II RB cells did not (*Figure 1D*). However, CD8SP cells from OT-II RB3 mice made Granzyme b and Perforin upon TCR stimulation (*Figure 1D*), demonstrating that CD8-lineage cells from OT-II RB3 mice are functionally cytotoxic-lineage cells.

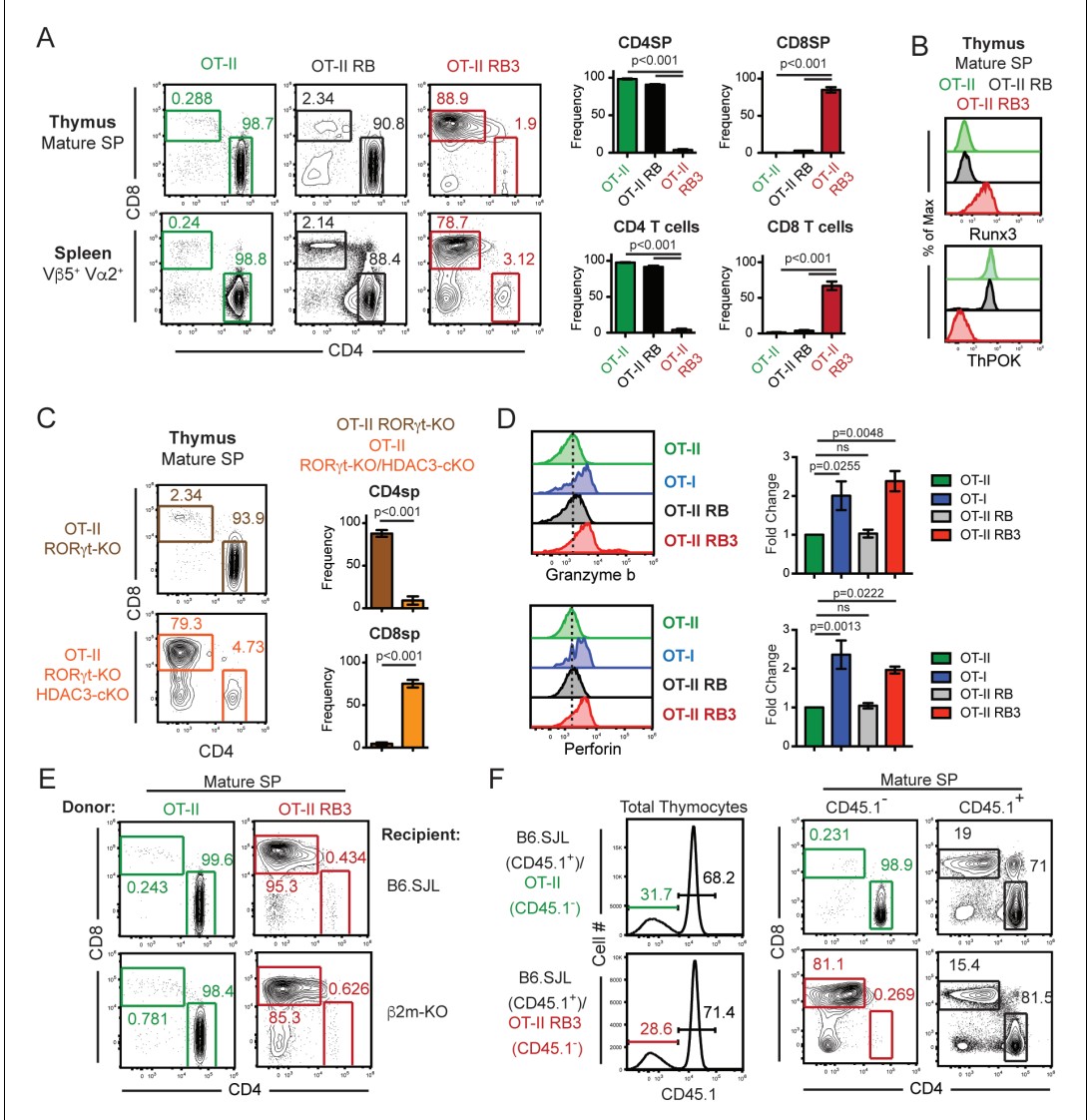

**Figure 1.** MHC class II restricted thymocytes are redirected to the CD8-lineage when HDAC3 is absent. (**A**) CD4/CD8 profile of mature thymic SP (Vβ5$^+$Vα2$^+$H2K$^+$CD24$^{lo}$) and splenic T cells (Vβ5$^+$Vα2$^+$) from OT-II, OT-II RORγt-KO Bcl-xl tg (RB) and OT-II RB3 mice. Graphs depict frequency of gated cells from at least three independent experiments (n = 3–5/group) (**B**) Runx3 and ThPOK expression in mature thymic SP cells from OT-II, OT-II RB, and OT-II RB3 mice (n = 4/group from four independent experiments). (**C**) CD4/CD8 profile of mature thymic SP from OT-II RORγt-KO and OT-II RORγt-KO HDAC3-cKO mice. Graphs depict frequency of gated cells from at least three independent experiments (n = 3–4/group). (**D**) Granzyme b and perforin expression in SP thymocytes from OT-II, OT-I, OT-II RB and OT-II RB3 mice. FACS plots were gated on CD4SP cells for OT-II and OT-II RB mice and CD8SP cells for OT-I and OT-II RB3 mice. Bar graphs depict mean ± SEM (n = 3/group from three independent experiments) of the fold change in MFI between unstimulated and TCR/CD2 stimulated conditions. (**E**) Representative FACS plots of the proportion of CD4SP and CD8SP mature thymocytes from straight BMCs, where bone marrow from OT-II or OT-II RB3 mice were transplanted into B6.SJL or β2m-KO recipients. Mature SP thymocytes were gated as Vβ5$^+$Vα2$^+$H2K$^+$CD24$^{lo}$. Mice (n = 3–5/group from three independent experiments) were analyzed 8–10 weeks after transfer. (**F**) Representative FACS plots of the proportion of CD4SP and CD8SP mature thymocytes from mixed BMCs from OT-II (CD45.1$^-$)/B6.SJL (CD45.1$^+$) and OT-II RB3 (CD45.1$^-$)/B6.SJL (CD45.1$^+$) mice. Mature SP thymocytes from CD45.1$^+$ cells were gated as H2K$^+$CD24$^{lo}$; mature SP thymocytes from CD45.1$^-$ cells were gated as Vβ5$^+$Vα2$^+$H2K$^+$CD24$^{lo}$. Mice were analyzed 8–10 weeks after transfer. (n = 5/group from three independent experiments).

DOI: https://doi.org/10.7554/eLife.43821.002

To exclude the contribution of TCRs selecting off of MHC class I in OT-II RB3 mice, bone marrow chimeras were generated where bone marrow from OT-II or OT-II RB3 mice were transferred into B6.SJL or β2m-KO recipients. Since the OT-II TCR is MHC class II-restricted, OT-II thymocytes differentiated to the CD4-lineage in β2m-KO recipients (*Figure 1E*). In addition, thymocytes from OT-II

RB3 mice also did not select off of MHC class I, as OT-II RB3-expressing CD8SP cells developed in β2m-KO recipients (*Figure 1E*). Therefore, CD8-lineage differentiation in OT-II RB3 mice is not due to aberrant selection off of MHC class I.

To determine whether redirection of MHC class II-restricted thymocytes to the CD8-lineage is cell-intrinsic in OT-II RB3 mice, mixed bone marrow chimeras were generated. A 50:50 mixture of congenic B6.SJL bone marrow cells were mixed with bone marrow cells from either OT-II or OT-II RB3 mice. As expected, B6.SJL thymocytes and OT-II cells developed normally, while OT-II RB3 thymocytes developed into mature CD8SP cells (*Figure 1F*), demonstrating that redirection of MHC class II-restricted thymocytes to the CD8-lineage in OT-II RB3 mice is cell-intrinsic and could not be compensated by the presence of WT B6.SJL cells.

## Acceleration of CD8-lineage commitment in HDAC3-deficient thymocytes

CD4/CD8-lineage choice is determined by the careful coordination between the lineage-determining transcription factors ThPOK and Runx3, where ThPOK promotes CD4-lineage commitment and Runx3 promotes CD8-lineage commitment (*Dave et al., 1998*; *Woolf et al., 2003*). These transcription factors antagonize each other's expression, while promoting the expression of CD4 or CD8-lineage genes (*Sakaguchi et al., 2015*; *Luckey et al., 2014*; *Egawa and Littman, 2008*; *Wildt et al., 2007*). To examine whether expression of these lineage-determining transcription factors was altered during lineage commitment in OT-II RB3 mice, we used a previously established gating scheme that outlines three main phases of CD4/CD8-lineage choice defined by CD69 and CCR7 expression ((*Kimura et al., 2016*), *Figure 2A*): *Phase 0* identifies pre-positive selection thymocytes (Stage 1, CD69⁻CCR7⁻), *Phase 1* identifies thymocytes that down-regulate the CD8 coreceptor to test for TCR signal persistence (Stages 2–4, CD69⁺CCR7⁻/lo/int, highlighted in orange), and *Phase 2* identifies thymocytes that have committed to either the CD4 or CD8 lineage (Stages 5–6, CD69⁻/⁺CCR7⁺). In addition to examining lineage choice in OT-II RB3 and OT-II RB mice, OT-I and OT-II mice were used as controls for normal CD8-lineage and CD4-lineage commitment, respectively. As shown previously (*Kimura et al., 2016*), MHC class I-restricted OT-I-expressing thymocytes decreased CD8α during Phase 1 and re-expressed CD8α during Phase 2, coinciding with CD8-lineage commitment (*Figure 2B*). In addition, Runx3 was induced during Phase 2 (*Figure 2B*), demonstrating that Runx3 is expressed when cells commit to the CD8-lineage. OT-II and OT-II RB thymocytes exhibited a consistent decrease in CD8α expression through Phases 1–2 and no

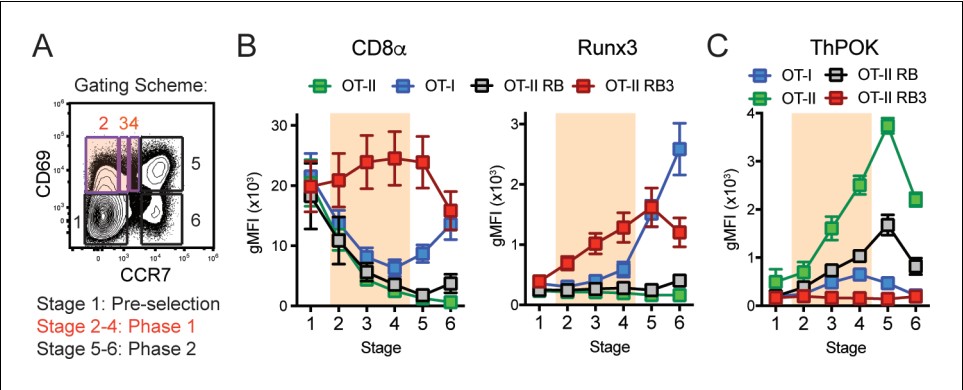

**Figure 2.** Acceleration of CD8-lineage commitment in HDAC3-deficient thymocytes. (**A**) Gating scheme outlining two phases of CD4/CD8-lineage commitment—Phase one includes CD69⁺ CCR7⁻ cells (Stage 2), CD69⁺ CCR7ˡᵒ cells (Stage 3), and CD69⁺ CCR7ⁱⁿᵗ cells (Stage 4); Phase two identifies CD69⁺ CCR7⁺ cells (Stage 5) and CD69⁻ CCR7⁺ cells (Stage 6). Orange shading highlights Phase 1. (**B**) Expression of Runx3 and CD8α during stages of lineage commitment (Stages 1–6), as outlined in (**A**) from OT-I, OT-II, OT-II RB and OT-II RB3 mice. CD69-versus-CCR7 plots were gated from DN-removed, CD45.2⁺ cells. (**C**) Expression of ThPOK during stages of lineage commitment, gated as in (**B**). Plots in B and C show mean ±SEM of MFI from 5 to 7 mice per group from five independent experiments.

DOI: https://doi.org/10.7554/eLife.43821.003

expression of Runx3 (*Figure 2B*), as expected for MHC class II-restricted thymocytes that commit to the CD4-lineage. However, OT-II RB3 mice showed premature expression of Runx3 during Phase 1 of lineage choice (*Figure 2B*). This pattern coincides with high CD8α expression during Phase 1 (*Figure 2B*). Therefore, deletion of HDAC3 leads to an acceleration of commitment to the CD8-lineage.

Since Runx3 is the lineage-determining transcription factor for the CD8-lineage and can directly antagonize ThPOK expression (*Kakugawa et al., 2017*), ThPOK expression was also examined during CD4/CD8-lineage commitment in OT-I, OT-II, OT-II RB and OT-II RB3 mice according to the same gating scheme used in *Figure 2A*. As expected, ThPOK was induced during Phase one in OT-II and OT-II RB mice and maintained at high levels in Phase 2, while in OT-I mice, ThPOK was expressed at low levels during Phase one but repressed at Phase 2 (*Figure 2C*). The downregulation of ThPOK at Phase two in OT-I mice corresponded with the induction of Runx3 at Phase 2 (*Figure 2B–C*), illustrating antagonism between Runx3 and ThPOK during lineage commitment. On the other hand, OT-II RB3 mice failed to induce ThPOK expression during Phase 1 (*Figure 2C*), which coincided with premature expression of Runx3 at this phase (*Figure 2B*). Altogether, the lower levels of ThPOK in OT-II RB3 mice suggest that as a consequence of acceleration of commitment to the CD8-lineage, factors that mediate CD4-lineage commitment are not induced after positive selection when HDAC3 is absent.

## HDAC3-deficient thymocytes fail to induce the CD4-lineage program

To investigate why HDAC3-deficient thymocytes prematurely commit to the CD8-lineage, RNA-seq and ChIP-seq was performed on thymocytes prior to CD4/CD8-lineage commitment. The two FACS-sorted populations analyzed were *Immature* ($V\beta5^{lo/-}CD69^-CD4^+CD8^+$) and *Selecting* ($V\beta5^{int}CD69^+$) cells (*Figure 3A*, *Figure 3—figure supplement 1A*) from OT-II and OT-II HDAC3-cKO mice. *Immature* and *Selecting* cells identify two developmental stages immediately upstream of CD4-lineage commitment in OT-II mice; *Immature* cells were DP thymocytes pre-positive selection (*Figure 3—figure supplement 1A*) and *Selecting* cells were cells that have received a TCR signal, illustrated by CD69 expression, but had not yet completed positive selection and upregulated TCRβ (*Figure 3—figure supplement 1B*). During the *Selecting* phase, thymocytes were also preparing for lineage choice, as they expressed markers that change between DP and $CD4^+CD8^{lo}$ thymocytes at an intermediary level (*Figure 3—figure supplement 1C*). In particular, IL-7R expression on *Selecting* cells was very low (*Figure 3—figure supplement 1C*), demonstrating that cells are not ready to undergo lineage choice because they are not yet cytokine receptive as $CD4^+CD8^{lo}$ thymocytes are. OT-II TCR transgenic mice were used to determine how HDAC3 deletion affects gene expression and chromatin specifically in MHC class II-restricted thymocytes, and OT-II HDAC3-cKO mice were used opposed to OT-II RB3 mice in order to examine transcriptomic and histone acetylation changes prior to positive selection.

Among the differentially expressed genes (defined by $Log_2$fold change of 1.5 and a false discovery rate (FDR) of <0.05), 449 genes were down-regulated and 164 genes were up-regulated in *Immature* (DP) cells from OT-II HDAC3-deficient cells compared to OT-II (*Figure 3—source data 1*). In *Selecting* ($CD69^+$) cells, 679 genes and 347 genes were down-regulated and up-regulated, respectively, from OT-II HDAC3-cKO mice compared to OT-II mice (*Figure 3—source data 1*). Alternatively, housekeeping gene expression did not change upon HDAC3 deletion (*Figure 3—figure supplement 2A*), indicating that deletion of HDAC3 did not change gene expression globally. Rather, there was evidence that CD4-lineage associated genes were down-regulated in OT-II HDAC3-cKO mice. In particular, *Zbtb7b* (encodes ThPOK) was markedly decreased in *Immature* (DP) and *Selecting* ($CD69^+$) cells compared to OT-II ($Log_2$ FC −2.38 and −3.83, respectively, *Figure 3—source data 1*). This led us to investigate whether genes corresponding to the CD4-lineage or the CD8-lineage were differentially regulated when HDAC3 was absent. To restrict our analysis to genes that distinguish between the CD4 and CD8-lineages, two gene sets were generated using the Immunological Genome Consortium Database (immgen.org, detailed in Materials and methods). To confirm specificity of the gene sets, the expression of these genes in DP, $CD4^+CD8^{lo}$, CD4SP and CD8SP thymocytes was examined. The CD4-lineage gene set showed low expression in DP and CD8SP thymocytes, induction at the $CD4^+CD8^{lo}$ stage, and high expression in CD4SP cells (*Figure 3B*). The CD8-lineage gene set was only expressed in CD8SP thymocytes (*Figure 3B*). Since expression of the CD4-lineage gene set started at the $CD4^+CD8^{lo}$ stage and neither gene sets were

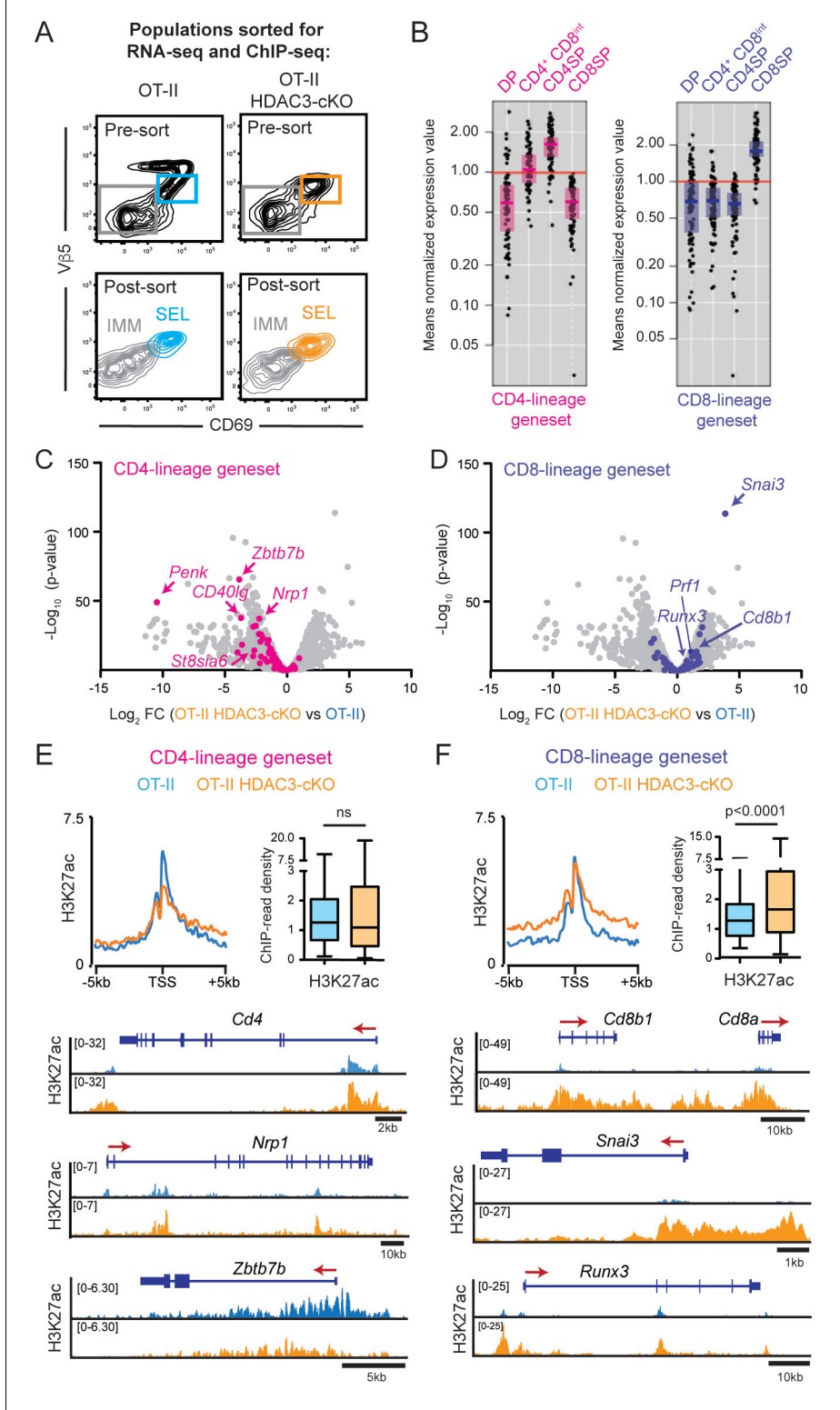

**Figure 3.** HDAC3-deficient thymocytes fail to induce the CD4-lineage program. (**A**) Flow cytometric analysis of populations from OT-II and OT-II HDAC3-cKO mice sorted for RNA-seq and ChIP-seq. Plots show representative pre-sort and post-sort analysis of two FACS sorted populations—*Immature* thymocytes (Vβ5$^{lo}$CD69$^-$ CD4$^+$CD8$^+$) and *Selecting* thymocytes (Vβ5$^{lo}$CD69$^+$). (**B**) Expression of CD4-lineage and CD8-lineage gene sets across DP, CD4$^+$CD8$^{lo}$, CD4SP, and CD8SP ImmGen expression data. (**C–D**) Volcano plot depicting the Log$_2$fold change (FC) between OT-II HDAC3-cKO and OT-II *Selecting* (CD69$^+$) cells. Grey dots show all genes; pink dots show CD4-

*Figure 3 continued on next page*

*Figure 3 continued*

lineage gene set (C); blue dots show CD8-lineage gene set (D). (E–F) Average H3K27ac ChIP-seq signal at the transcription start site (TSS) between OT-II and OT-II HDAC3-cKO *Selecting* (CD69$^+$) cells for CD4-lineage gene sets (E) and CD8-lineage gene sets (F). Box-and-whisker plots depict H3K27ac signal at the TSS at CD4- or CD8-lineage genes between OT-II and OT-II HDAC3-cKO mice. Snapshots of example ChIP-seq tracks for each gene set are below. See also *Figure 3—figure supplements 1–5* and *Figure 3—source datas 1–2*.
DOI: https://doi.org/10.7554/eLife.43821.004

The following source data and figure supplements are available for figure 3:

**Source data 1.** Gene expression values (RNA-seq) of *Immature* and *Selecting* populations from OT-II and OT-II HDAC3-cKO mice.
DOI: https://doi.org/10.7554/eLife.43821.010

**Source data 2.** Gene lists: CD4-lineage, CD8-lineage, silenced genes, housekeeping genes.
DOI: https://doi.org/10.7554/eLife.43821.011

**Figure supplement 1.** Characterization of *Immature* and *Selecting* thymic populations.
DOI: https://doi.org/10.7554/eLife.43821.005

**Figure supplement 2.** Gene expression of select housekeeping genes, CD4-lineage genes, and CD8-lineage genes.
DOI: https://doi.org/10.7554/eLife.43821.006

**Figure supplement 3.** Heatmaps of CD4-lineage and CD8-lineage genes.
DOI: https://doi.org/10.7554/eLife.43821.007

**Figure supplement 4.** H3K9ac signal at CD4-lineage and CD8-lineage genes.
DOI: https://doi.org/10.7554/eLife.43821.008

**Figure supplement 5.** H3K9ac signal and H3K27ac signal at silenced genes and housekeeping genes in thymocytes.
DOI: https://doi.org/10.7554/eLife.43821.009

---

expressed at the DP stage, we examined how the CD4-lineage gene set was differentially expressed at the *Selecting* (CD69$^+$) stage between OT-II and OT-II HDAC3-cKO mice. 64% of CD4-lineage genes in this gene set were significantly down-regulated in HDAC3-deficient cells (*Figure 3C*). Examples include *CD4*, *Zbtb7b* (ThPOK), *Itgb3*, *Nrp1*, *Cd40lg*, and *St8sia6* (*Figure 3C*, *Figure 3—figure supplements 2B* and *3A*). However, 53% of CD8-lineage genes were significantly up-regulated in OT-II HDAC3-cKO cells compared to OT-II (*Figure 3D*), such as *Snai3*, *Cd8a*, *Cd8b1*, *Prf1*, *Itgae*, *Gpr114*, and *Runx3* (*Figure 3—figure supplements 2C* and *3B*). Protein expression showed a similar pattern, as there was a decrease in ThPOK expression and slight increase in Runx3 expression at the *Selecting* (CD69$^+$) stage in OT-II HDAC3-cKO mice compared to OT-II mice (*Figure 3—figure supplement 2D*). Of note, this pattern was also observed in OT-II RB3 mice, where there was an increase in Runx3 and decrease in ThPOK protein expression early in lineage choice compared to OT-II cells (*Figure 2B–C*), illustrating convergence between OT-II HDAC3-cKO mice and OT-II RB3 mice. This distinct pattern suggests that lack of HDAC3 leads to a gene expression pattern predisposed to the CD8-lineage and against the CD4-lineage prior to positive selection.

To determine whether the different patterns of gene expression between OT-II and OT-II HDAC3-cKO mice at the *Selecting* (CD69$^+$) stage was reflective of changes in histone acetylation, the level of H3K27ac and H3K9ac around the transcription start site (TSS) was examined at CD4-lineage and CD8-lineage genes. Interestingly, there were minimal changes in the average H3K27ac signal (*Figure 3E*) or average H3K9ac signal (*Figure 3—figure supplement 4A*) at CD4-lineage genes. This was either due to the lack of change in acetylation between OT-II and OT-II HDAC3-cKO cells (e.g. *Cd4*, *Nrp1*, *Figure 3E*, *Figure 3—figure supplement 4A*) or a decrease in H3K9ac or H3K27ac signal (e.g. *Zbtb7b*, *Figure 3E*, *Figure 3—figure supplement 4A*). However, there was a significant increase in the average H3K27ac signal and average H3K9ac signal around the TSS of CD8-lineage genes (e.g. *Cd8a*, *Cd8b1*, *Snai3*, *Runx3*, *Figure 3F*, *Figure 3—figure supplement 4B*). Consistent with the RNA-seq data, genes that are normally silenced in thymocytes (i.e. macrophage genes, B-cell genes, HSC/progenitor genes) or housekeeping genes (i.e. mitochondrial genes, ribosomal genes) showed no change in histone acetylation between OT-II and OT-II HDAC3-cKO thymocytes (*Figure 3—figure supplement 5A–B*). Altogether, this shows that HDAC3-deficient thymocytes induce CD8-lineage genes and fail to prompt the CD4-lineage program after receiving a TCR signal.

## DP thymocytes from HDAC3-cKO mice show a CD8-lineage bias at the chromatin level

Regulation of cell identity occurs at the chromatin level through *cis* and *trans* elements, such as enhancers. Super-enhancers are areas of the genome that control genes important for cell type specification (*Hnisz et al., 2013*; *Whyte et al., 2013*). Therefore, alterations in super-enhancers may play a role in the re-direction of MHC class II-restricted cells to the CD8-lineage in HDAC3-deficient cells. Very high levels of H3K27ac reliably characterize super-enhancers (*Hnisz et al., 2013*). Therefore, the H3K27ac ChIP-seq dataset was used to identify super-enhancers in *Immature* (DP) cells from OT-II and OT-II HDAC3-cKO mice to determine whether prior to receiving a TCR signal, HDAC3-deficient DP thymocytes show a CD8-lineage bias. Using the ROSE (Rank Ordering of Super-Enhancers) algorithm to identify super-enhancers (*Whyte et al., 2013*; *Lovén et al., 2013*) within the *Immature* (DP) compartment, 5967 typical-enhancers were identified from OT-II mice and 7496 typical enhancers from OT-II HDAC3-cKO mice, while 263 super-enhancers were identified from OT-II mice and 360 super-enhancers from OT-II HDAC3-cKO mice (*Figure 4A*), demonstrating that deletion of HDAC3 leads to an increase in the number of enhancers. Further examination revealed a significant increase in H3K27ac signal at super-enhancers from HDAC3-deficient cells compared to WT (*Figure 4B–C*), while the average H3K27ac signal at typical enhancers remained largely unchanged (*Figure 4B*). Not only was the H3K27ac signal increased at super enhancers in HDAC3-cKO cells but the median length was also increased compared to the OT-II mice (*Figure 4D*). In addition, the increase in acetylation at super-enhancers from OT-II HDAC3-cKO mice corresponded to an increase in gene expression of genes associated with super-enhancers, compared to OT-II mice (*Figure 4E*). To determine whether the increase in super-enhancer number in HDAC3-deficient *Immature* (DP) cells was due to the acquisition of new super-enhancers, the number of overlapping super-enhancers between OT-II and OT-II HDAC3-cKO mice was examined. 197 super-enhancers overlapped between the two mice, 66 super-enhancers were unique to OT-II mice, and 163 super-enhancers were unique to OT-II HDAC3-cKO mice (*Figure 4F*). Upon further examination of the 66 unique super-enhancers from OT-II mice, the level of H3K27ac at these regions was comparable between OT-II and OT-II HDAC3-cKO mice (*Figure 4G*), as well as the expression of genes associated with those super enhancers (*Figure 4H*). The H3K27ac signal SE cutoff for OT-II mice was lower than the SE cutoff for OT-II HDAC3-cKO mice (~10,000 versus~35,000, respectively), suggesting that the 'unique' super-enhancers in OT-II mice were similarly acetylated between the groups even though they did not reach the threshold for super enhancers in OT-II HDAC3-cKO mice. Alternatively, the unique super-enhancers from OT-II HDAC3-cKO mice exhibited an increase in H3K27 acetylation signal compared to the same regions in OT-II mice (*Figure 4I*). Among the 163 unique super-enhancers in OT-II HDAC3-cKO mice, super-enhancers near *Runx3* and *Patz1* (encodes MAZR) were especially notable (*Figure 4—source data 1*). Runx3 and MAZR have been shown to promote CD8-lineage commitment (*Sakaguchi et al., 2015*; *Kohu et al., 2005*; *Woolf et al., 2003*), therefore the formation of *Runx3* and *Patz1* super-enhancers suggests that HDAC3-deficient DP thymocytes are biased for the CD8-lineage prior to positive selection and lineage choice. Interestingly, the super-enhancer regions near *Runx3* from OT-II HDAC3-cKO mice were identified as multiple typical enhancers in OT-II mice (*Figure 4J*), demonstrating that the lack of HDAC3 leads to an increase in acetylation at enhancer regions near *Runx3*, stitching of enhancers together, and categorizing the region as a super-enhancer. In fact, this increase in H3K27ac signal corresponds with an increase in mRNA expression (*Figure 4K*), which is a feature of super-enhancers (*Whyte et al., 2013*). In sum, deletion of HDAC3 leads to a bias for the CD8-lineage at the *Immature* (DP) stage by quantitatively increasing CD8-specific enhancer function.

## HDAC3-deficient DP thymocytes show elevated IL-21R signaling and STAT5 activation

Common gamma chain cytokine signaling drives CD8-lineage commitment under normal conditions (*Park et al., 2010*). Since pre-selection thymocytes show a transcriptomic and chromatin bias for the CD8-lineage when HDAC3 is absent, cytokine signaling may also be altered in HDAC3-deficient DP thymocytes. Therefore, the expression of common gamma chain receptors were examined in DP thymocytes between WT and HDAC3-cKO mice. In WT mice, DP thymocytes did not express IL-7Rα, IL-15Rα, and IL-2Rβ, and expressed common gamma chain (γc), IL-4Rα, and IL-21R at low levels

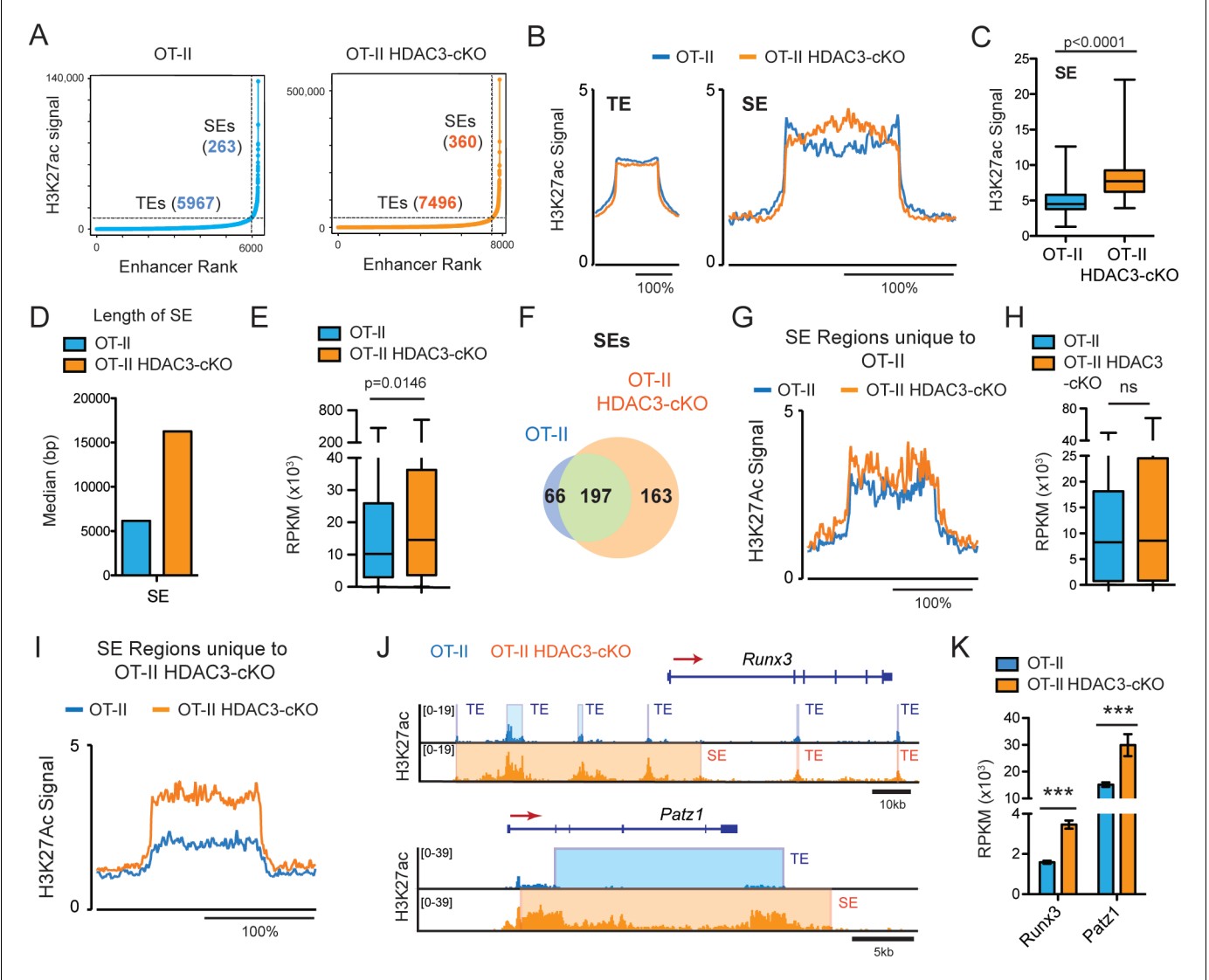

**Figure 4.** DP thymocytes from HDAC3-cKO mice show a CD8-lineage bias at the chromatin level. (**A**) Typical enhancers (TEs) and super enhancers (SEs) in *Immature* (DP) cells from OT-II and OT-II HDAC3-cKO mice. (**B**) Average H3K27ac signal (normalized to input) for TE and SE regions from OT-II and OT-II HDAC3-cKO mice. (**C**) Box-and-whisker plot of H3K27ac signal for each super enhancer in OT-II and OT-II HDAC3-cKO mice. (**D**) Median length of super-enhancers between OT-II and OT-II HDAC3-cKO mice. (**E**) Box-and-whisker plots of mRNA expression (reads per kilobase of exon per million mapped reads, RPKM) from super-enhancer-associated genes from OT-II and OT-II HDAC3-cKO mice. (**F**) Venn Diagram of shared and unique SEs between OT-II and OT-II HDAC3-cKO mice. (**G**) H3K27ac signal at super-enhancer regions unique to OT-II mice and the corresponding regions in OT-II HDAC3-cKO mice. (**H**) Box-and-whisker plots of mRNA expression (RPKM) between OT-II and OT-II HDAC3-cKO mice of genes associated with super-enhancers unique to OT-II mice. (**I**) H3K27ac signal at super-enhancer regions unique to OT-II HDAC3-cKO mice and the corresponding regions in OT-II mice. (**J**) Snapshot of H3K27ac ChIP-seq tracks at the *Runx3* and *Patz1* locus from OT-II (blue) and OT-II HDAC3-cKO (orange) mice. Shaded regions depict TEs and SEs. (**K**) Gene expression (RNA-seq) of Runx3 and Patz1 in *Immature* (DP) cells from OT-II and OT-II HDAC3-cKO mice. (**B, G, I**) The x-axis represents a surrounding area that corresponds to 200% of the center of each region. (***, p < 0.001). See also *Figure 4—source data 1*.
DOI: https://doi.org/10.7554/eLife.43821.012

The following source data is available for figure 4:

**Source data 1.** Super-enhancer list.
DOI: https://doi.org/10.7554/eLife.43821.013

compared to isotype control (**Figure 5A**). Similarly, DP thymocytes from HDAC3-cKO mice did not express IL-7Rα, IL-15Rα, and IL-2Rβ (**Figure 5A**), however common gamma chain (γc), IL-4Rα, and IL-21R expression were significantly increased compared to WT mice (**Figure 5A**). Likewise, DP thymocytes from OT-II RB3 mice exhibited a significant increase in IL-21R expression compared to DP

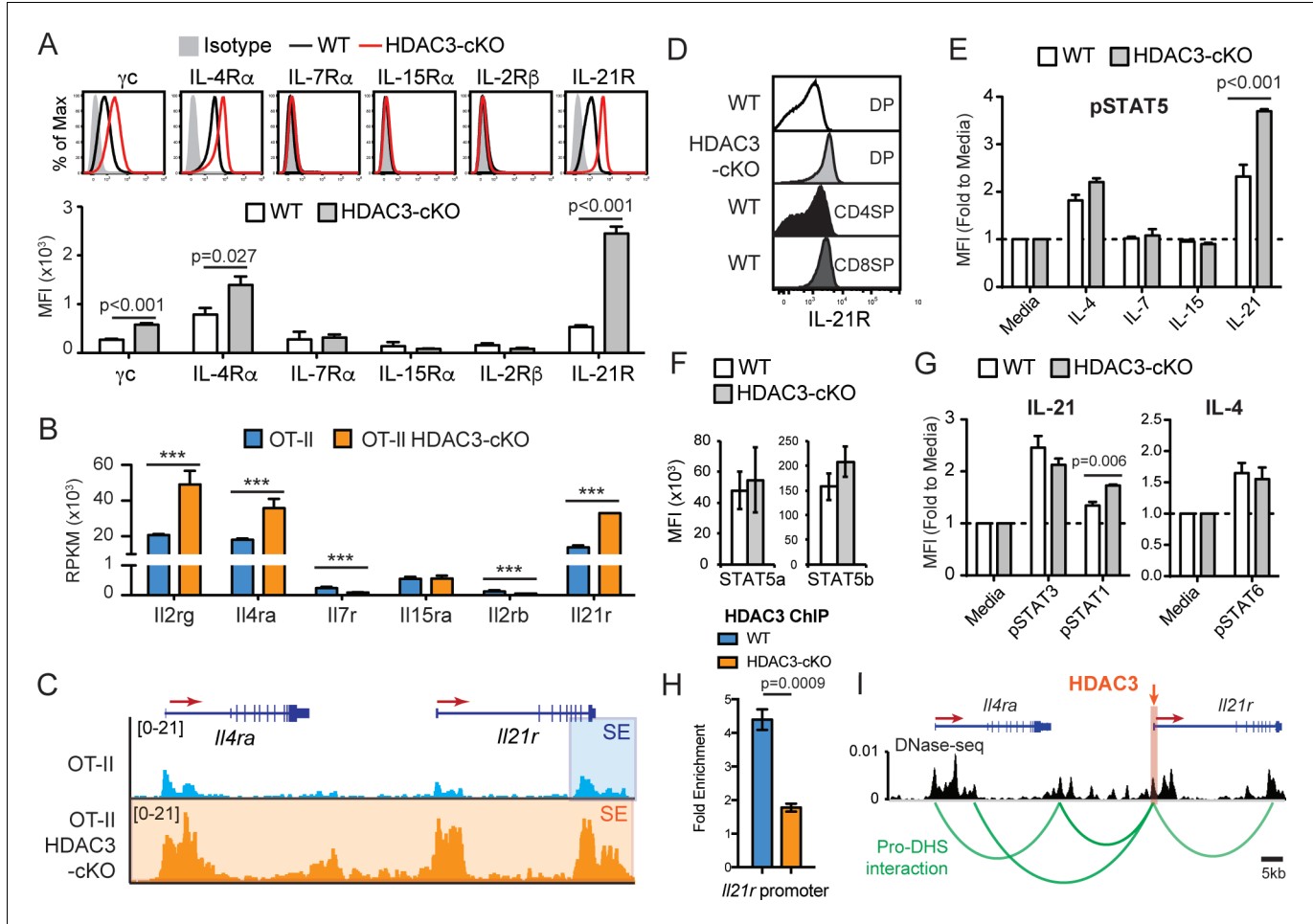

**Figure 5.** DP thymocytes from HDAC3-cKO mice show increased STAT5 activation in response to IL-21. (**A**) Surface expression of cytokine receptor chains γc, IL-4Rα, IL-7Rα, IL-15Rα, IL-2Rβ, and IL-21R on DP thymocytes from WT and HDAC3-cKO mice. Flow cytometric plots have an isotype control to illustrate background level of expression. (**B**) Gene expression (RNA-seq) of *Il2rg*, *Il4ra*, *Il7r*, *Il15ra*, *Il2rb*, and *Il21r* in *Immature* (DP) cells from OT-II and OT-II HDAC3-cKO mice. (**C**) Snapshot of H3K27ac ChIP-seq tracks at the *Il4ra* and *Il21r* gene loci in *Immature* (DP) thymocytes from OT-II and OT-II HDAC3-cKO mice. Shaded regions identify super-enhancers. (**D**) Protein expression of IL-21R on DP, CD4SP and CD8SP thymocytes from WT mice and DP thymocytes from HDAC3-cKO mice. (**E**) pSTAT5 levels in DP thymocytes from WT and HDAC3-cKO mice after 10 min ex vivo stimulation with IL-21, IL-15, IL-7, IL4, or media alone. (**F**) Protein expression of STAT5a and STAT5b in DP thymocytes from WT and HDAC3-cKO mice. (**G**) pSTAT3 and pSTAT1 levels after a 10 min in vitro IL-21 stimulation, and pSTAT6 levels after IL-4 stimulation of DP thymocytes from WT and HDAC3-cKO mice. (**H**) Quantitative ChIP (qChIP) of HDAC3 binding at the *Il21r* promoter in DP thymocytes from WT and HDAC3-cKO mice. Graph depicts fold enrichment over *Rpl30* (n = 4 mice/group from four indpendent experiments). (**I**) DNase-seq and Hi-C arc plots at the *Il4ra* and *Il21r* gene loci in DP thymocytes and pooled DN3-to-DP thymocytes, respectively. Shaded region highlights where HDAC3 binds, as shown in **Figure 5H**. (DHS, DNA hypersensitivity sites). (**A, E, G**) Bar graph shows mean ± SEM of MFI from 4 to 5 mice from at least three independent experiments. (**D, H**) Plots are representative of at least three mice from three independent experiments. (**F**) Bar graph shows mean ± SEM of MFI from three mice from two independent experiments. (\*\*\*, p < 0.001). See also **Figure 5—figure supplements 1–2**.

DOI: https://doi.org/10.7554/eLife.43821.014

The following figure supplements are available for figure 5:

**Figure supplement 1.** IL-21R expression and signaling in OT-II RB3 mice.
DOI: https://doi.org/10.7554/eLife.43821.015
**Figure supplement 2.** IL-21R expression and signaling in CD2-icre HDAC3-cKO mice.
DOI: https://doi.org/10.7554/eLife.43821.016

thymocytes from OT-II, OT-I, and OT-II RB mice (*Figure 5—figure supplement 1A*). In addition, mRNA levels of *Il2rg*, *Il4ra,* and *Il21r* were higher in OT-II HDAC3-deficient DP thymocytes compared to OT-II DP thymocytes (*Figure 5B*). Furthermore, the *Il4ra* and *Il21r* gene loci were hyperacetylated in OT-II HDAC3-deficient DP thymocytes compared to OT-II DP thymocytes (*Figure 5C*). Overall, there is a clear up-regulation of select common gamma chain receptors in HDAC3-deficient cells, and, interestingly, the elevated levels of IL-21R observed on DP thymocytes from HDAC3-cKO mice were comparable to IL-21R expression on CD8SP thymocytes from WT mice (*Figure 5D*), suggesting that deletion of HDAC3 leads to a pre-mature upregulation of IL-21R on DP thymocytes.

Typically, WT DP thymocytes are resistant to STAT5 activation due to low cytokine receptor expression and high SOCS1 expression (*Yu et al., 2006*). However, IL-4 and IL-21 receptor expression was markedly increased in HDAC3-cKO mice. To determine whether stimulation by IL-4 or IL-21, as well as other common gamma chain-containing receptors, leads to STAT5 activation in DP thymocytes, STAT5 phosphorylation (p-STAT5) was measured after stimulation with IL-4, IL-7, IL-15, or IL-21. There was no change in p-STAT5 levels after IL-7 or IL-15 stimulation of thymocytes from WT or HDAC3-cKO mice, as expected since DP thymocytes lack IL-7R and IL-15R expression (*Figure 5E*). Alternatively, IL-21 stimulation led to a significant increase in p-STAT5 in HDAC3-deficient thymocytes, with about a 2-fold increase over WT cells and a 4-fold increase over no stimulation (*Figure 5E*). Similarly, DP thymocytes from OT-II RB3 mice exhibited a significant increase in p-STAT5 after IL-21 stimulation compared to DP thymocytes from OT-II, OT-I, and OT-II RB mice (*Figure 5—figure supplement 1B*). This increase in p-STAT5 in HDAC3-deficient thymocytes was not due to a change in total STAT5a or STAT5b levels (*Figure 5F*). To determine whether the increased IL-21R expression in HDAC3-deficient DP thymocytes also leads to enhanced activation of STAT3 and STAT1, p-STAT3 and p-STAT1 levels were measured after IL-21 stimulation. There was no change in p-STAT3 levels between WT and HDAC3-cKO mice after IL-21 stimulation, while p-STAT1 levels showed a slight increase in HDAC3-cKO mice compared to WT mice (*Figure 5G*). Similarly, there was no change in p-STAT6 levels between WT and HDAC3-cKO mice after IL-4 stimulation (*Figure 5G*). Therefore, in DP thymocytes from HDAC3-cKO mice, STAT5 activation was enhanced in response to IL-21 stimulation.

To determine whether IL-21R expression and signaling occurs prior to the DP stage in HDAC3-cKO mice, we examined IL-21R expression and pSTAT5 signaling in DN and immature SP (ISP; CD8$^+$CD4$^-$TCRβ$^-$) thymocytes from WT and HDAC3-cKO mice. IL-21R expression was similar in DN thymocytes between mice, however ISPs from HDAC3-cKO mice exhibited a significant increase in IL-21R expression compared to WT mice (*Figure 5—figure supplement 2A*). Of note, the level of IL-21R expression in ISPs from HDAC3-cKO mice was less than what was observed in DP thymocytes (*Figure 5—figure supplement 2A*). In addition, upon IL-21 stimulation, ISPs from HDAC3-cKO mice exhibited a significant increase in phospho-STAT5 levels compare to WT mice, while stimulation with IL-4, IL-7, and IL-15 showed no effect (*Figure 5—figure supplement 2B*). Therefore, IL-21R expression and signaling precedes the DP stage.

A major function of HDAC3 is to repress gene expression. To determine whether HDAC3 associates with the *Il21r* gene locus in DP thymocytes to regulate IL-21R expression, HDAC3 binding was measured by quantitative ChIP (qChIP). Interestingly, HDAC3 was enriched at the *Il21r* promoter in DP thymocytes from WT mice compared to HDAC3-cKO mice (*Figure 5H*), demonstrating that HDAC3 interacts with the *Il21r* promoter directly. The *Il21r* gene locus resides next to *Il4ra* on chromosome 7 in mice. As shown previously, deletion of HDAC3 results in elevated histone acetylation at both *IL21r* and *Il4ra* gene loci (*Figure 5C*). Interestingly, the increase in acetylation in HDAC3-deficient cells resulted in a dramatic expansion of a super-enhancer detected near the transcription end site of *Il21r* in OT-II cells (*Figure 5C*), suggesting regulation of *Il21r* and *Il4ra* may be coordinated. Publicly available DNase-seq and Hi-C of WT DP thymocytes were examined for chromatin looping between regulatory elements and gene promoters within the *Il21r* and *Il4ra* gene loci (dataset from *Hu et al., 2018*). Remarkably, the *Il21r* promoter looped with multiple regulatory elements (DNase-seq peaks) at the *Il4ra* gene locus (*Figure 5I*), suggesting these genes are co-regulated. Since HDAC3 associates with the *Il21r* promoter, which interacts with the *Il4ra* locus through chromatin looping, HDAC3 may also function to reduce histone acetylation at *Il4ra*. Altogether, HDAC3 is required to reduce expression of the IL-21R and IL-4R cytokine receptors in DP thymocytes.

## Blocking IL-21R does not restore CD4-lineage commitment in OT-II RB3 mice

HDAC3 is required to restrain cytokine signaling in DP thymocytes. Since cytokine signaling drives CD8-lineage commitment and HDAC3-deficient thymocytes exhibit accelerated commitment to the CD8-lineage, elevated IL-21R signaling may promote the redirection of MHC class II-restricted thymocytes to the CD8-lineage when HDAC3 is absent. To test this, OT-II and OT-II RB3 mice were treated with an anti-IL-21R blocking antibody and assessed for CD4-lineage commitment. After two weeks of antibody injections, the CD4/CD8 profile was analyzed within the mature SP thymocyte compartment. As expected, isotype and anti-IL-21R antibody treated OT-II mice produced CD4SP thymocytes (*Figure 6A*). However, blocking IL-21R signaling in OT-II RB3 mice did not restore CD4-lineage commitment, as both isotype and anti-IL-21R antibody treatments produced CD8SP thymocytes in OT-II RB3 mice (*Figure 6A*). To determine whether blocking IL-21R signaling altered ThPOK or Runx3 expression early in CD4/CD8-lineage choice, OT-II and OT-II RB3 mice were examined for expression of these transcription factors in Vβ5$^{int}$H2K$^{neg}$ thymocytes (see *Figure 6—figure supplement 1* for gating). As expected, there was similar ThPOK expression early in CD4/CD8-lineage choice between isotype and anti-IL-21R antibody-treated OT-II mice (*Figure 6B*). However, OT-II RB3 mice treated with an anti-IL-21R blocking antibody failed to induce ThPOK (*Figure 6B*). In addition, Runx3 expression was still induced early in lineage choice in both isotype and anti-IL-21R blocking antibody treated OT-II RB3 mice (*Figure 6C*). Therefore, blocking IL-21R signaling in HDAC3-deficient thymocytes is not sufficient to correct for the redirection of MHC class II-restricted thymocytes to the CD8-lineage.

## HDAC3 binds to *Runx3* and *Patz1* regulatory elements in DP and CD4SP thymocytes

Since blocking IL-21R signaling could not alter the aberrant lineage commitment in HDAC3-deficient thymocytes, HDAC3 may function to repress the expression of CD8-lineage genes directly. To understand the mechanisms of HDAC3-mediated *Runx3* and *Patz1* gene regulation, HDAC3 binding was measured at the *Runx3* and *Patz1* gene loci in DP thymocytes. Since HDAC3 deletion results in

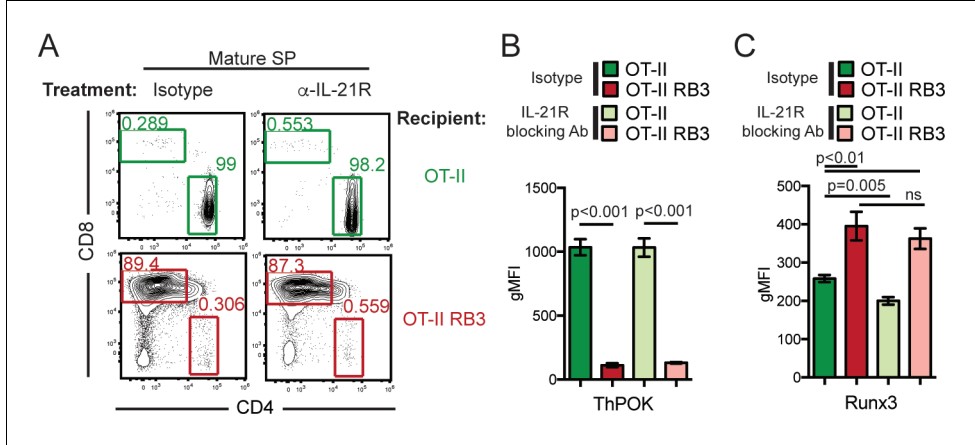

**Figure 6.** Blocking IL-21R does not restore CD4-lineage commitment in OT-II RB3 mice. (**A**) OT-II and OT-II RB3 mice were injected with isotype or anti-IL-21R blocking antibodies retro-orbitally (i.v.) for two weeks every 3–4 days. Flow plots depict CD4/CD8 profile in mature SP thymocytes (Vβ5$^+$Vα2$^+$H2K$^+$CD24$^{lo}$). Data is representative of 3–4 mice per group from two independent experiments. (**B–C**) ThPOK (**B**) and Runx3 (**C**) expression in Vβ5$^{hi}$H2K$^-$ thymocytes from OT-II and OT-II RB3 mice treated with isotype or anti-IL-21R blocking antibody, as performed in A. Data show mean ± SEM of geometric MFI from 3 to 4 mice from two independent experiments. See also *Figure 6—figure supplement 1*.

DOI: https://doi.org/10.7554/eLife.43821.017

The following figure supplement is available for figure 6:

**Figure supplement 1.** Gating strategy for *Figure 6B–C*.
DOI: https://doi.org/10.7554/eLife.43821.018

the formation of super enhancers near *Runx3* and *Patz1* prior to positive selection (*Figure 4J*), HDAC3 may associate with these regions to regulate histone acetylation. Candidate HDAC3 binding sites were determined from examination of publicly available HDAC3 ChIP-seq datasets (*Nanou et al., 2017*; *Wang et al., 2009*). Datasets from pro-B cells and human CD4 T cells revealed HDAC3 binding at a region 50 kb upstream of *Runx3*, the *Runx3* promoter, and within exon 1 of *Patz1* (*Figure 7—figure supplement 1*). In particular, these binding sites correspond with areas where super enhancers form in HDAC3-deficient DP thymocytes (*Figure 7—figure supplement 1*). Likewise, in DP thymocytes, HDAC3 bound to the region 50 kb upstream of *Runx3* and exon 1 of *Patz1*, however HDAC3 did not bind to the *Runx3* promoter (*Figure 7A*). Since the HDAC3 binding site is 50 kb upstream of *Runx3*, the DNA-seq and Hi-C datasets were revisited to examine for chromatin looping and whether HDAC3 interacts with the *Runx3* gene locus in DP thymocytes (*Hu et al., 2018*). In fact, the *Runx3* promoter associated with four regulatory elements upstream, including one near where HDAC3 binds (*Figure 7B*). Therefore, HDAC3 binds to regulatory elements

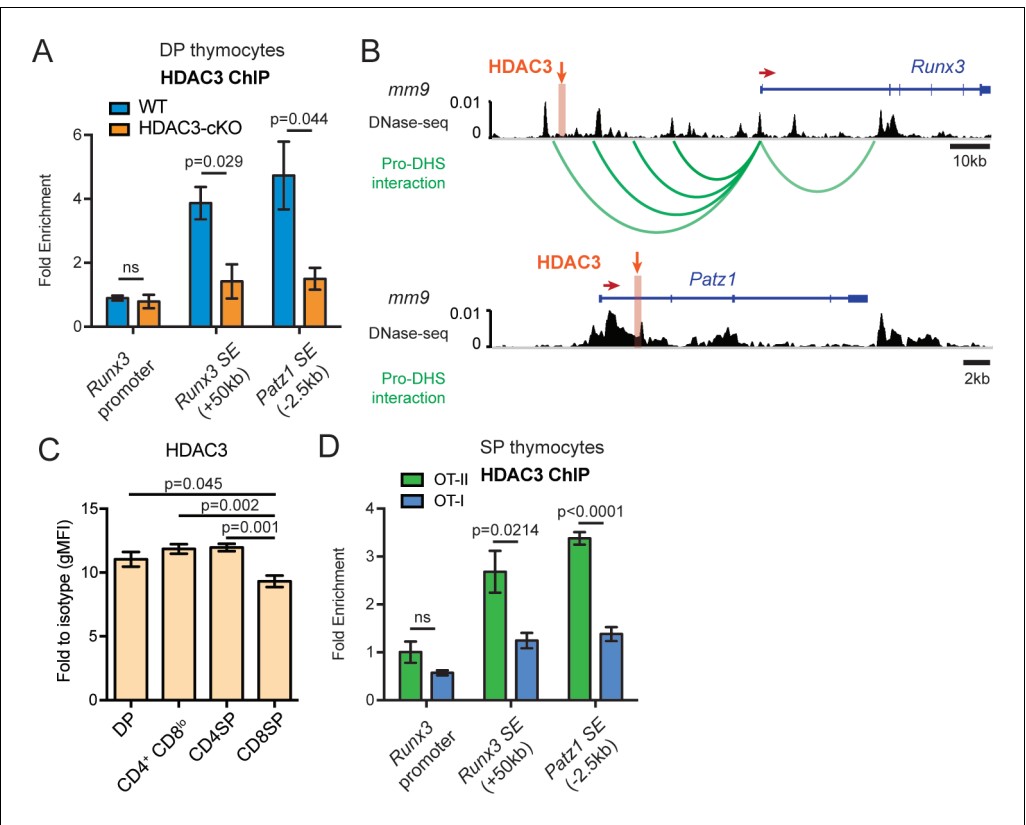

**Figure 7.** HDAC3 binds to *Runx3* and *Patz1* regulatory regions in DP and CD4SP thymocytes. (**A**) HDAC3 qChIP in DP thymocytes from WT and HDAC3-cKO mice. Plots show mean ± SEM of fold enrichment over *Rpl30* (n = 3 mice/group from three independent experiments). (**B**) DNase-seq and Hi-C arc plots at the *Runx3* and *Patz1* gene loci in DP thymocytes and pooled DN3-to-DP thymocytes, respectively. Shaded region highlights where HDAC3 binds, as shown in (**A**). (DHS, DNA hypersensitivity sites). (**C**) HDAC3 protein expression measured by flow cytometry in DP, CD4+CD8lo, CD4SP, and CD8SP thymocytes from WT mice. Plots show mean ± SEM of fold enrichment over isotype (n = 4 mice/group from three independent experiments). (**D**) HDAC3 qChIP in SP thymocytes from OT-II and OT-I mice. Plots show mean ± SEM of fold enrichment over *Rpl30* (n = 4 mice/group from two independent experiments). See also *Figure 7—figure supplements 1–2*.

DOI: https://doi.org/10.7554/eLife.43821.019

The following figure supplements are available for figure 7:

**Figure supplement 1.** HDAC3 ChIP-seq in pro-B cells and human CD4 T cells.
DOI: https://doi.org/10.7554/eLife.43821.020
**Figure supplement 2.** Model.
DOI: https://doi.org/10.7554/eLife.43821.021

associated with *Runx3* and *Patz1*, suggesting that HDAC3 has a direct regulatory role in repressing CD8-lineage promoting genes in DP thymocytes.

Since HDAC3 is required to repress *Runx3* and *Patz1* expression in DP thymocytes, HDAC3 may also regulate *Runx3* and *Patz1* expression differently between CD4SP and CD8SP thymocytes. First, HDAC3 protein expression was measured during T cell development by flow cytometry to determine whether HDAC3 protein is regulated differently during lineage choice. Interestingly, HDAC3 protein expression was significantly reduced in CD8SP thymocytes compared to DP, CD4⁺CD8ˡᵒ, and CD4SP thymocytes (*Figure 7C*). In fact, HDAC3 qChIP revealed that HDAC3 binding to *Runx3* and *Patz1* regions was lost in CD8SP thymocytes and maintained in CD4SP thymocytes (*Figure 7D*). This suggests a direct regulatory role of HDAC3 to repress *Runx3* and *Patz1* in DP and CD4SP thymocytes and removal of HDAC3 from *Runx3* and *Patz1* loci is important for CD8-lineage commitment.

## Discussion

The ability to undergo error-free CD4/CD8-lineage choice is critical for maintaining the adaptive immune system. Here, we demonstrate that the histone deacetylase HDAC3 is required for CD4-lineage commitment. Two mouse models were used to study this, OT-II RB3 mice and OT-II HDAC3-cKO mice. OT-II RB3 mice, the mouse model that bypassed the positive selection defect in HDAC3-cKO mice, exhibited redirection of MHC class II-restricted cells to the CD8-lineage and accelerated kinetics towards CD8-lineage commitment. OT-II HDAC3-cKO mice, which have a block in positive selection, revealed *Immature* (DP) thymocytes to be biased for the CD8-lineage prior to positive selection and *Selecting* (CD69⁺) cells failed to initiate the CD4-lineage program. We determined that HDAC3 binds to the *Il21r* promoter and is required to control IL-21R expression and STAT5 activation in DP thymocytes, however blocking IL-21R signaling did not restore CD4-lineage commitment in OT-II RB3 mice. Rather, HDAC3 binds to the CD8-lineage promoting genes *Runx3* and *Patz1* to restrain CD8-lineage genes for CD4-lineage commitment.

Regulation of cytokine signaling is critical for proper T cell development. In particular, cytokine signaling is restrained in DP thymocytes and drives CD8-lineage commitment after positive selection. 75% of MHC class I-restricted thymocytes undergo CD8-lineage commitment via γc cytokine receptor signaling, primarily through IL-7 (*McCaughtry et al., 2012*; *Etzensperger et al., 2017*). IL-7R signaling is shut off in DP thymocytes, as enforced IL-7R signaling results in the CD8-lineage fate (*Park et al., 2010*). Whether overexpression of STAT5-inducing cytokine receptors in DP cells other than IL-7R can induce CD8-lineage commitment was unknown. Here, we show that HDAC3 deletion leads to increased IL-21R expression in DP thymocytes. IL-21 is secreted by thymic epithelial cells, therefore overactive IL-21R signaling in DP thymocytes may promote CD8-lineage commitment. However, blocking IL-21R signaling did not restore CD4-lineage choice in HDAC3-deficient thymocytes.

Thymocytes navigate through distinct steps to undergo error-free lineage choice. Recently, Kimura *et al* identified two phases of lineage commitment in MHC class I-restricted thymocytes—phase one identifies cells testing for TCR signal persistence, and phase two classifies cells receiving cytokine signaling that results in CD8-lineage commitment (*Kimura et al., 2016*). Expression of ThPOK and Runx3 corresponds with these two phases, as ThPOK is induced in phase 1and Runx3 in phase 2 (*Kimura et al., 2016*). Cytokine signaling is confined to phase 2 and drives Runx3 expression, as SOCS1 transgenic OT-I mice failed to induce Runx3 expression in phase 2 (*Kimura et al., 2016*). However, this study focused on MHC class I-restricted cells. In our study, we found that MHC class II-restricted thymocytes erroneously committed to the CD8-lineage. Using the same gating scheme to distinguish the two phases of lineage commitment, Runx3 was expressed pre-maturely in phase 1 in OT-II RB3 mice. Correspondingly, ThPOK was not induced in OT-II RB3 mice, suggesting that a true 'phase 1' may not have occurred and commitment to the CD8-lineage was accelerated. Consistent with this, CD8α was not downregulated in OT-II RB3 mice at phase 1. We show an epigenetic mechanism where HDAC3 is required to restrain Runx3 expression in DP thymocytes to maintain a bipotential state. In DP thymocytes, HDAC3 binds to a regulatory element upstream of *Runx3*, and deletion of HDAC3 results in hyperacetylation at this locus, as well as an increase in *Runx3* mRNA levels. Therefore, HDAC3 functions to restrain CD8-lineage promoting genes for CD4-lineage development.

While class I HDACs play a role in regulating CD4/CD8-lineage identity, our study identifies a distinct role for HDAC3 to prevent redirection of MHC class II-restricted cells to the CD8-lineage. Combined deletion of HDAC1 and HDAC2 via CD4-cre results in loss of CD4-lineage integrity, as peripheral CD4 helper T cells aberrantly express the CD8 coreceptor (CD4+CD8+ T cells) as well as the CD8 effector T cell program upon activation (*Boucheron et al., 2014a*). It appears that CD4/CD8-lineage commitment is unaffected in these mice, but rather HDAC1/2 function to constitutively repress CD8-lineage genes in CD4 T cells in the periphery (*Boucheron et al., 2014b*). Alternatively, deletion of HDAC3 by CD2-icre results in a transcriptomic and histone acetylation bias for the CD8-lineage prior to positive selection and misdirection of thymocytes to the CD8-lineage. In fact, HDAC3 binds to *Runx3* and *Patz1* regulatory elements in DP and CD4 SP thymocytes in WT mice. However, CD4-cre mediated deletion of HDAC3 resulted in no defect in CD4/CD8-lineage integrity (*Hsu et al., 2015*), as HDAC3 protein is still present at the DP stage in these mice (*Philips et al., 2016*). This demonstrates that HDAC3 is specifically required for restraining CD8-lineage genes is DP thymocytes, however redundant mechanisms in addition to HDAC3 are required for maintenance of CD8-lineage gene repression in CD4SP thymocytes. Compared to HDAC1/2 maintaining CD4-lineage identity in the periphery, HDAC3 functions to limit CD8-lineage genes in DP thymocytes prior to CD4-lineage commitment.

## Materials and methods

### Mice

*Hdac3* fl/fl mice (*Knutson et al., 2008*) mice were previously described. B6.SJL, Bcl-xL Tg mice (*Chao et al., 1995*), RORγt-knockout (KO) mice (*Ivanov et al., 2006*), and CD2-icre mice (*Zhumabekov et al., 1995*), OT-II mice (*Barnden et al., 1998*), and OT-I mice (*Hogquist et al., 1994*) were purchased from The Jackson Laboratory. Mice were housed in barrier facilities and experiments were performed at the Mayo Clinic with the approval of the Institutional Animal Care and Use Committee. All mice were analyzed between the ages of 3 and 12 weeks. All genetically modified mice were examined with either littermate or age-matched controls, which may include floxed-only mice (no Cre), CD2-icre, RORγt-het, or WT mice, as no differences were observed between these mice. For convenience, the control mice in each experiment are termed wild-type (WT) but may represent either floxed-only, CD2-icre, RORγt-het or WT mice. No difference was observed between male and female mice. Mouse genotypes were confirmed by flow cytometry and/or by PCR after euthanizing the mouse.

### Bone Marrow Chimeras

Mixed bone marrow chimeras (BMCs) were generated by i.v. injecting $4 \times 10^6$ cells from either a 50:50 mix of OT-II (CD45.1$^{-/-}$)/B6.SJL (CD45.1$^{+/+}$) or OT-II RB3 (CD45.1$^{-/-}$)/B6.SJL (CD45.1$^{+/+}$) mice into lethally irradiated congenic B6.SJL (CD45.1$^{+/+}$) recipients. In experiments where OT-II RB3 straight chimeras were used, $4 \times 10^6$ bone marrow cells from OT-II and OT-II RB3 mice were i.v. injected into lethally irradiated congenic B6.SJL (CD45.1$^{+/+}$) or β2m-KO (CD45.1$^{-/-}$) recipients. Antibodies recognizing the OT-II TCR transgene (Vβ5$^+$Vα2$^+$) were used to exclude the contribution of hematopoietic cells from CD45.1$^-$ β2m-KO recipients. Recipient mice from mixed BMCs or straight chimeras received the antibiotic enrofloxacin in their drinking water for 3 weeks and were analyzed after 8 weeks.

### Flow cytometry

FACS analysis was performed on an Attune NxT flow cytometer (Thermo Fisher), and all experiments were analyzed using FlowJo software (v9.5). Cytoplasmic and nuclear proteins were examined via intracellular flow cytometry. Thymocytes or splenic lymphocytes were labeled with surface markers before being fixed and permeabilized with a Foxp3/Transcription Factor Staining Buffer Set (for nuclear protein staining; eBioscience or Tonbo Biosciences), True-Nuclear Transcription Factor Buffer Set (for SOCS1 or SOCS3 staining; Biolegend) or an intracellular fixation and permeabilization buffer kit (for cytoplasmic protein staining; BD Biosciences). All analyses included size exclusion (forward scatter [FSC] area/side scatter [SSC] area), doublet exclusion (FSC height/FSC area), and dead cell exclusion (fixable viability dye; eBioscience or Tonbo Biosciences). All other reagents for flow

cytometry were purchased from BD Biosciences, eBioscience, BioLegend, Tonbo Biosciences, Cell Signaling, or Abcam. Unless stated in other Materials and methods section, the following clones for flow cytometry were used: CD4 (clone RM4-5), CD8α (clone 53–6.7), TCR Vβ5.1/5.2 (clone MR9-4), TCR Vα2 (clone B20.1), Runx3 (clone R3-5G4), ThPOK (clone T43-94), Granzyme B (clone NGZB), CD24 (clone M1/69), H-2K$^b$ (clone AF6-88.5), CCR7 (clone 4B12), CD132 (common γ chain) (clone TUGm2), IL-4Rα (clone I015F8), IL-7Rα (clone A7R34), IL-15Rα (clone DNT15Ra), CD122 (IL-2Rβ) (clone 5H4), IL-21R (clone 4A9), ROR gamma (t) (clone B2D), Bcl-xl (clone 7B2.5), CD45.2 (clone 104), HDAC3 (ab7030).

## Stimulations

To examine phosphorylation of STAT proteins, total thymocytes were stimulated with 20 ng/mL of IL-4, 10 ng/mL of IL-7, 10 ng/mL of IL-15, or 10 ng/mL of IL-21 or left unstimulated for 10 min in complete culture media (RPMI 1640 with 10% FCS, L-glutamine, penicillin, and streptomycin). After stimulation, cells were immediately fixed with BD Lyse/Fix Buffer (BD Phosflow kit), stained for surface proteins, permeabolized with BD Perm Buffer III (BD Phosflow kit), and stained with antibodies targeting pSTAT5 (#9365, Cell Signaling Technology), pSTAT1 (#8009, Cell Signaling Technology), pSTAT3 (#4324, Cell Signaling Technology), or pSTAT6 (#9361, Cell Signaling Technology) for 30 min at room temperature. Secondary antibody (#4414, Cell Signaling Technology) was used for pSTAT6 and stained for 10 min on ice. To examine Granzyme b and Perforin after stimulation, total thymocytes were left unstimulated or stimulated with plate bound 5 μg/mL of anti-TCRβ (H57-597; BioXCell) and 5 μg/mL of soluble anti-CD2 (RM2-5; eBioscience). After two days of culture, cells were harvested and stained for flow cytometry.

## FACS sorting

Cell sorting for ChIP-seq and RNA-seq was performed on the BD FACS Aria (BD Biosciences). Total thymocytes were stained with CD4 (RM4-5), CD8α (53–6.7), Vβ5 (B21.5), CD69 (H1.2F3), and DAPI. *Immature* thymocytes were gated as Live Vβ5$^{lo}$ CD69$^-$ CD4$^+$ CD8α$^+$. *Selecting* thymocytes were gated as Live Vβ5$^{int}$ CD69$^+$. Post-sorting analysis was performed to verify purity. For RNA-seq, individual samples were sorted and used in biological triplicate. For ChIP-seq, individual samples were sorted, pooled from 4 OT-II mice or 6 OT-II HDAC3-cKO mice, and then separated into equal samples for IP (see ChIP-seq for details).

## RNA-seq and analysis

RNA was extracted from FACS sorted cells with Qiazol (Qiagen) and purified using the RNeasy Micro kit (Qiagen) using DNase treatment, according to the manufacturer's protocol. 200 ng of total RNA was used to make libraries using Illumina TruSeq RNA Library Preparation Kit v2 (Illumina) according to the manufacturer's protocol. RNA-seq libraries were sequenced on the Illumina HiSeq 4000 platform at Medical Genome Facility at Mayo Clinic. Gene expression analysis was performed using the Mayo RNA-sequencing analysis pipeline, MAP-RSeq (*Kalari et al., 2014*). Briefly, FASTQ reads (paired-end; 100 bp) were aligned to mm10 using TopHat2 (*Trapnell et al., 2009*), raw gene and exon counts are generated by featureCounts (*Liao et al., 2014*), and the quality of samples was determined using RSeQC (*Wang et al., 2012*). edgeR v3.8.6 (*Robinson et al., 2010*) was used to perform the differential expression analysis comparing the several groups among each other. Genes which had a False discovery rate (FDR) < 0.05 and an absolute log$_2$Fold Change > 1.5 were considered to be significantly differentially expressed.

## ChIP-seq

ChIP-seq was performed as previously described (*Zhong et al., 2017*). Sorted cells were cross-linked with 1% formaldehyde (final concentration) for 10 min and quenched with 125 mM glycine for 5 min at room temperature. After washing with TBS, cells were resuspended in cell lysis buffer (10 mM Tris-HCl, pH7.5, 10 mM NaCl, 0.5% NP-40) and incubated on ice for 10 min. Lysates were washed with MNase digestion buffer (20 mM Tris-HCl, pH7.5, 15 mM NaCl, 60 mM KCl, 1 mM CaCl$_2$). After resuspension in 200 μL fresh MNase digestion buffer containing a proteinase inhibitor cocktail (Sigma), the lysates were incubated with 2000 units of MNase (NEB, Cat # M0247S) per 4 × 10$^6$ cells at 37°C for 20 min with continuous mixing in thermal mixer. After adding 200 μL of sonication buffer

(100 mM Tris-HCl, pH8.1, 20 mM EDTA, 200 mM NaCl, 2% Triton X-100, 0.2% sodium deoxycholate), the lysates were sonicated for 30 min (30 s on/30 s off) using Bioruptor Twin (UCD-400) (Diagenode, Inc.) and centrifuged at 21,130 x g for 10 min. The supernatants were collected, and the chromatin content was estimated by the Qubit assay (Invitrogen). For normalization of ChIP efficiency, *Drosophila* chromatin equivalent to about 5% of total chromatin was added. The chromatin was then incubated with anti-H3K27ac antibody (Abcam, Cat.# ab4729, lot# GR150367) or in-house generated anti-H3K9ac antibody (EDL lot 1) on a rocker overnight. Protein G-magnetic beads (30 µL, Life Technologies) were added, and further incubated for 3 hr. The beads were extensively washed with ChIP buffer (50 mM Tris-HCl, pH8.1, 10 mM EDTA, 100 mM NaCl, 1% Triton X-100, 0.1% sodium deoxycholate), high salt buffer (50 mM Tris-HCl, pH8.1, 10 mM EDTA, 500 mM NaCl, 1% Triton X-100, 0.1% sodium deoxycholate), LiCl$_2$ buffer (10 mM Tris-HCl, pH8.0, 0.25 M LiCl$_2$, 0.5% NP-40, 0.5% sodium deoxycholate, 1 mM EDTA), and TE buffer. Bound chromatin was eluted and reverse-crosslinked at 65°C overnight. DNAs were purified using Min-Elute PCR purification kit (Qiagen) after the treatment of RNase A and proteinase K. ChIP-seq libraries were prepared from about 5 ng ChIP and input DNA using the ThruPLEX DNA-seq Kit V2 (Rubicon Genomics, Ann Arbor, MI). ChIP enrichment was validated in library DNAs by performing real-time PCR (see *Supplementary file 1* for primer sequences). The libraries were sequenced to 51 base pairs from both ends on an Illumina HiSeq 2000 instrument in the Mayo Clinic Center for Individualized Medicine Medical Genomics Facility.

## ChIP-seq analysis

ChIP-seq data was processed using the HiChIP pipeline (*Yan et al., 2014*) incorporating ChIP-Rx normalization (*Orlando et al., 2014*). Briefly, mouse reads were aligned to mm10 using Burrows-Wheeler Aligner (BWA) (*Li et al., 2009*) and Drosophila reads aligned to dm3 using custom scripts. Picard (http://broadinstitute.github.io/picard/) was used to mark duplicates. Read pairs without a unique alignment were filtered out using SAMTools (*Li and Durbin, 2009*) and a custom script that only retains pairs with one or both ends uniquely mapped. Mouse reads were normalized to Drosophila reads according to *Orlando et al., 2014*. Peaks were called using MACS2 (*Zhang et al., 2008*) for H3K27ac and H3K9ac. The ROSE software package (*Whyte et al., 2013*; *Lovén et al., 2013*) was used for typical enhancer and super enhancer analysis. Data was visualized using Integrative Genomics Viewer (IGV) and EaSeq (*Lerdrup et al., 2016*).

## Defining genesets

Genesets for RNA-seq and ChIP-seq analysis were generated using Immgen.org (Immunological Genome Consortium; *Heng et al., 2008*). Specifically, we created CD4-lineage and CD8-lineage genesets using the 'Population Comparison' program, which performs a comparison of gene expression between different immune populations according to fold change, p-value, and FDR. CD4SP ('T_4SP24-_Th') and CD8SP ('T_8SP24-_Th') populations were compared using Microarry V1 dataset provided by the software. The CD4-lineage geneset consists of the top 100 genes expressed in the CD4SP population, and the CD8-lineage geneset consists of the top 100 genes expressed in the CD8SP population. To confirm the gene lists corresponded with the appropriate thymic developmental stages, the 'My GeneSet' program was used to determine how each geneset is expressed in DP ('T_DP_Th'), CD4 +CD8 lo ('T_4 + 8int_Th'), CD4SP ('T_4SP24-_Th') and CD8SP ('T_8SP24-_Th') populations. The plot generated by 'My GeneSet' program was used in *Figure 3B*. See *Figure 3— source data 2* for gene list. For RNA-seq data analysis using CD4-lineage and CD8-lineage gene sets, genes that did not have transcripts were excluded from the gene list.

## IL-21R blocking experiments

OT-II and OT-II RB3 mice were retro-orbitally (i.v.) injected with 100 µg of isotype (polyclonal Armenian hamster IgG, BioXCell) or anti-IL-21R (4A9, BioXCell) antibody for two weeks every 2–3 days. After the last injection, thymocytes were harvested and processed for flow cytometry.

## Quantitative ChIP (qChIP)

For HDAC3 qChIP, DP thymocytes were enriched using the EasySep Mouse Streptavidin Rapid-Spheres Isolation kit (#19860, StemCell Technologies) to remove SP and DN thymocytes with biotin-

conjugated anti-CCR7 (4B12), anti-IL-7Rα (A7R34), anti-H2K (AF6-88.5), anti-CD44 (IM7), and anti-CD25 (PC61.5) as well as anti-TCRγδ (UC7-13D5) antibodies. For SP thymocyte enrichment, Mouse PE Positive Selection kit (#18554, StemCell Technologies) was used positively select for CCR7+ thymocytes (PE-conjugated anti-CCR7 (4B12)). ChIP was performed according to *Pchelintsev et al. (2016)*, with the following adjustments: After cell lysis and brief sonication (four mins, 30 s on/30 s off; Bioruptor Pico (Diagenode, Inc.)), equal volume of 2x MNase buffer (35 mM Tris-HCl pH 7.5, 25 mM NaCl, 120 mM KCl, 2 mM CaCl$_2$) was added to each sonication sample along with 3.75 U/μl of MNase (Cell Signaling Technologies #10011S) at 37°C for 15 mins. Anti-HDAC3 antibody (#85057, Cell Signaling Technologies) was used to isolate chromatin bound to HDAC3 after chromatin fragment size was confirmed to be less than 500 bp by gel electrophoresis. Isolated DNA was used to perform real-time PCR. Graphs depict fold enrichment over regions without HDAC3 binding (*Rpl30* primers, Cell Signaling Technologies (#7015)). See *Supplementary file 1* for primer sequences.

## Statistical analysis

To compare between groups, unpaired student t test was used to compare frequency of mature SP thymocytes, Granzyme b/Perforin expression, H3K27ac signal at super-enhancer regions, mRNA levels of gene associated with super-enhancers, cytokine receptor protein expression, and HDAC3 binding via qChIP. The exactTest (edgeR software) was used to compare mRNA levels (RPKM) at individual genes. Student t test was calculated using Microsoft Excel or GraphPad Prism.

## Accession numbers

In house generated ChIP-seq and RNA-seq were deposited to GEO with accession number GSE109531. Publicly available datasets were retrieved from GEO series: GSM2195840 (thymus DNase-seq; mm10), GSM2096648 (pro-B cell HDAC3 ChIP-seq; mm10), GSM393952 (human CD4 T cell HDAC3 ChIP-seq; hg18), GSE79422 (pooled DN3-to-DP Hi-C and associated DNase-seq; mm9). ChIP-seq and DNase-seq datasets were visualized by IGV (mm9). The Hi-C dataset was visualized with the online WashU epigenome browser session (mm9) at http://epigenomegateway.wustl.edu/browser/?genome=mm9&session=bxT0F5m0YY, provided by *Hu et al. (2018)*.

## Acknowledgements

We thank Dr. Scott Hiebert for HDAC3 floxed mice. This work benefitted from data assembled by the Immunological Genome Consortium. We thank members of the Epigenomics Program and Medical Genome Facility within the Center for Individualized Medicine for their contributions to the generation of ChIP-seq and RNA-seq datasets. We also thank members of the VSS and Kay Medina (Mayo Clinic) laboratory for thoughtful discussions and critical reading of the manuscript. This work was supported by National Institutes of Health Grants R56 AI122746 and T32 AI007425, Center for Biomedical Discovery at Mayo Clinic, Mayo Graduate School funds to RLP, and Mayo Foundation funds to VSS. The authors do not have any conflicts of interest.

## Additional information

### Funding

| Funder | Grant reference number | Author |
|---|---|---|
| National Institute of Allergy and Infectious Diseases | R56 AI122746 | Virginia Smith Shapiro |
| Mayo Clinic | Center for Biomedical Discovery | Virginia Smith Shapiro |
| Mayo Clinic | Graduate School | Rachael Laura Philips |
| Mayo Foundation | | Virginia Smith Shapiro |
| National Institute of Allergy and Infectious Diseases | T32 AI007425 | Rachael Laura Philips |

The funders had no role in study design, data collection and interpretation, or the decision to submit the work for publication.

## Author contributions

Rachael Laura Philips, Conceptualization, Data curation, Formal analysis, Validation, Investigation, Visualization, Methodology, Writing—original draft, Writing—review and editing; Jeong-Heon Lee, Investigation, Writing—review and editing; Krutika Gaonkar, Pritha Chanana, Data curation, Software; Ji Young Chung, Sinibaldo R Romero Arocha, Aaron Schwab, Resources, Formal analysis, Writing—review and editing; Tamas Ordog, Methodology, Writing—review and editing; Virginia Smith Shapiro, Conceptualization, Resources, Supervision, Funding acquisition, Validation, Methodology, Project administration, Writing—review and editing

## Author ORCIDs

Virginia Smith Shapiro http://orcid.org/0000-0001-9978-341X

## Ethics

Animal experimentation: This study was performed in strict accordance with the recommendations in the Guide for the Care and Use of Laboratory Animals of the National Institutes of Health. All of the animals were handled according to approved institutional animal care and use committee (IACUC) protocol (#A1984-16) of the Mayo Clinic.

## Decision letter and Author response

Decision letter https://doi.org/10.7554/eLife.43821.037
Author response https://doi.org/10.7554/eLife.43821.038

# Additional files

## Supplementary files

• Supplementary file 1. Primers used for ChIP-seq and qChIP.
DOI: https://doi.org/10.7554/eLife.43821.022

• Transparent reporting form
DOI: https://doi.org/10.7554/eLife.43821.023

## Data availability

Sequencing data have been deposited in GEO under accession code GSE109531. Source data has been uploaded for Figures 3 and 4 (Excel files).

The following dataset was generated:

| Author(s) | Year | Dataset title | Dataset URL | Database and Identifier |
|---|---|---|---|---|
| Rachael Laura Philips, Jeong-Heon Lee, Krutika Gaonkar | 2018 | HDAC3 restrains CD8-lineage genes to maintain a bi-potential state in CD4+CD8+ thymocytes for CD4-lineage commitment | http://www.ncbi.nlm.nih.gov/geo/query/acc.cgi?acc=GSE109531 | NCBI Gene Expression Omnibus, GSE109531 |

The following previously published datasets were used:

| Author(s) | Year | Dataset title | Dataset URL | Database and Identifier |
|---|---|---|---|---|
| ENCODE DCC | 2016 | DNase-seq from thymus (ENCLB504VIQ) | https://www.ncbi.nlm.nih.gov/geo/query/acc.cgi?acc=GSM2195840 | NCBI Gene Expression Omnibus, GSM2195840 |
| Aikaterini Nanou | 2016 | ChIP_HDAC3minus 1 | https://www.ncbi.nlm.nih.gov/geo/query/acc.cgi?acc=GSM2096648 | NCBI Gene Expression Omnibus, GSM2096648 |
| Chongzhi Zang | 2009 | CD4-HDAC3 | https://www.ncbi.nlm. | NCBI Gene |

| | | | nih.gov/geo/query/acc. cgi?acc=GSM393952 | Expression Omnibus, GSM393952 |
|---|---|---|---|---|
| Gangqing Hu | 2018 | Integrative analysis of 3D nucleome and chromatin accessibility reveals a chromatin barrier established for T-lineage commitment during early T cell development [Dnase-Seq, HiC-Seq, Mnase-Seq, RNA-Seq] | https://www.ncbi.nlm. nih.gov/geo/query/acc. cgi?acc=GSE79422 | NCBI Gene Expression Omnibus, GSE79422 |
| Liang Yang | 2009 | Immunological Genome Project data Phase 1 | https://www.ncbi.nlm. nih.gov/geo/query/acc. cgi?acc=GSE15907 | NCBI Gene Expression Omnibus, GSE15907 |

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
