## [Decision Letter]

[Editors’ note: in a previous version of this study, the authors were asked to provide a plan for revisions before the editors issued a final decision. What follows is the editors’ letter requesting such plan.]

Thank you for submitting your article "HDAC3 restrains cytokine signaling in CD4^+^CD8^+^ thymocytes for the generation of CD4 T cells" for consideration by *eLife*. Your article has been reviewed by three peer reviewers, one of whom is a member of our Board of Reviewing Editors, and the evaluation has been overseen by Tadatsugu Taniguchi as the Senior Editor. The reviewers have opted to remain anonymous.

The reviewers have discussed the reviews with one another and the Reviewing Editor has drafted this decision to help you prepare a revised submission. As you will note, there are substantial issues to be addressed and we like to know whether you will be able to submit the revised version within two months.

At this point, we request that you respond with a detailed plan of action and a time table for the completion of this work. The editor will then consider your responses, possibly in consultation with the reviewers, and get back to you with advice.

Summary:

Philips et al. explore the role of HDAC3 in CD4/CD8 lineage commitment and show that HDAC3 loss causes a striking redirection of MHC class II-restricted TCR-transgenic DP thymocytes to the CD8 fate. HDAC3-deficient mice have defects in intrathymic selection, previously studied by these workers. In this work, the authors correct for defects in RORγT down-regulation using mice deficient for RORγT and correct for the survival defect resulting from RORγT deficiency by introducing a transgene for Bcl-xl. Such mice were previously shown to produce only CD8 but not CD4 cells, although subsequent investigation in the manuscript largely use a simpler OT-II TCR-transgenic HDAC3-deficient model, and thymocytes that initiate TCR signaling are isolated and studied. The authors attempt to unravel the molecular basis for this highly penetrant CD4/CD8 cell fate phenotype, revealing changes in the expression of cytokine receptors and in responsiveness to cytokine signaling. They find that HDAC3 deficiency leads to increased H3K27Ac deposition specifically on genes associated with the CD8 lineage, including *Runx3*, which is critical for commitment to the CD8 lineage. Surprisingly, enhanced Runx3 expression is shown to be secondary to IL-21 signaling. Finally, the authors use an antibody blocking strategy to demonstrate the IL-21R signaling is responsible for changes in acetylation and STAT5 binding the Runx3 locus in HDAC3-deficient mice, since antibody blockade of IL-21R reverses the changes in H3K27Ac, including the increased H3K27Ac at the *Il21r* locus. It is thus concluded that HDAC3-deficiency regulates lineage commitment through effects on cytokine signaling, mostly likely *Il21r*; specifically, that HDAC3 is critical to restrain cytokine signaling in thymocytes, and in its absence IL-21 signaling leads to STAT5 activation, and so CD8 lineage commitment.

Essential revisions:

While the manuscript by Philips et al. show data supporting an interesting role for HDAC3 in regulating the genomic preparedness for thymocyte lineage differentiation into the CD4 branch, the data do not yet quite provide for the visualisation of a robust mechanistic pathway, and there are a number of more formal concerns as well. If these limitations are addressed substantively, it would increase enthusiasm for publication in *eLife*.

1) How HDAC3 restrains IL-21R cytokine signaling remains unresolved. Also, does IL-21R expression and signaling precede the immature-DP super-enhancer alterations? Does the role of IL-21R in regulating genome-wide acetylation patterns indicate that acetylation is not driven directly by HDAC3? And instead, that HDAC3 has a limited role, for this function, in regulating IL-21R expression and/or signaling alone? The mechanistic model appears to be HDAC3-IL-21R-STAT5-PIAS. Is PIAS in fact essential for this specific functional role of HDAC3? Evidence for these mechanistic steps in mediating the final functional outcome of lineage choice would be very useful. A cartoon of the mechanistic pathways involved would also be very helpful.

2) Another serious omission is that the authors do not appear to address whether blockage of IL-21R blocks CD8 development in OTII-RB3 mice. While authors have demonstrated some effects of HDAC3 loss on cytokine signaling, they have not definitively demonstrated that this is the cause of altered lineage fate, as they have made no attempt to normalize cytokine signaling in HDAC3 mice and link this to restoration of CD4 lineage commitment. If the effect of HDAC3 on IL-21R expression and responsiveness is thought to play a key role, then it should be possible to extend the IL-21R blockade and examine effects on lineage commitment.

3) The functional role of HDAC3 is revealed primarily in RORγT-null Bcl-xl-transgenic mice. While the argument for this complex genotypic background is reasonable, it leaves open an uncertainty about the role of HDAC3 in wild-type double-positive (DP) thymocyte differentiation. Even limited functional data supporting such a role in functional assays in closer-to-wild-type mice would be useful.

4) The lack of inclusion of a Rag1/2-null genotype in the OT-II experiments (Figure 1) leaves the formal possibility open that endogenous MHC class I-restricted TCRs have provided positive selection into the CD8 lineage. Data to rule out such a possibility would be helpful.

5) The ChIP-seq and RNA-seq data in Figure 2 have been done on background genotypes (OT-II and OT-II+HDCA3-null) that are different from the ones used in Figure 1. It would be useful to provide data showing convergence between these various background genotypes.

6) In a number of instances, comparisons are offered between OT-II and OT-II-RB3 mice. It would be advisable to include OT-II-RB genotypes in these comparisons, to ensure formally that the major genotypic distinction is indeed in HDAC3.

[Editors’ note: the revision plan was rejected after consideration by the editor and reviewers, but the authors submitted for reconsideration. The decision letter issued after considering the revision plan is shown below.]

Thank you for submitting your article "HDAC3 restrains cytokine signaling in CD4^+^CD8^+^ thymocytes for the generation of CD4 T cells" for consideration by *eLife*. Your article has been reviewed by three peer reviewers, one of whom is a member of our Board of Reviewing Editors, and the evaluation has been overseen by Tadatsugu Taniguchi as the Senior Editor. The reviewers have opted to remain anonymous.

The reviewers had discussed the reviews with one another and the Reviewing Editor had drafted a letter to express our concerns and elicit a response to the reservations we had about this work. At that point, you will recollect that we sent you that letter and requested that you respond with a detailed plan of action and a time table for the completion of this work.

We thank you for your response to that request. The editor and reviewers have now discussed your proposed plans and consider them satisfactory. However, the *eLife* editorial policy is that manuscripts are to be rejected if revisions are expected to take more than two months. Therefore, while this is a formal rejection, you are strongly encouraged to submit the manuscript again as a new one after completion of the revision plan. Every effort will be made to ensure that such a submission will be evaluated by the same editors and reviewers. The individual reviews are provided below for your reference.

*Reviewer #1:*

The manuscript by Philips et al. show data supporting an interesting role for HDAC3 in regulating the genomic preparedness for thymocyte lineage differentiation into the CD4 branch. These findings may provide a potentially rich source of future investigation points into this functionally crucial stage of differentiation, although as they stand, the data do not yet quite provide for the visualisation of a robust mechanistic pathway, which remains something of a limitation. Further, significant concerns regarding formal rigour also remain. If these limitations are addressed substantively, it would increase enthusiasm for publication in *eLife*.

1) The functional role of HDAC3 is revealed primarily in RORγT-null Bcl-xl-transgenic mice. While the argument for this complex genotypic background is reasonable, it leaves open an uncertainty about the role of HDAC3 in wild-type double-positive (DP) thymocyte differentiation. Even limited functional data supporting such a role in functional assays in closer-to-wild-type mice would be useful.

2) The lack of inclusion of a Rag1/2-null genotype in the OT-II experiments (Figure 1) leaves the formal possibility open that endogenous MHC class I-restricted TCRs have provided positive selection into the CD8 lineage. Data to rule out such a possibility would be helpful.

3) Does the alteration in identified super-enhancers in pre-selection immature-DP thymocytes from HDAC-3-null OT-II mice continue into the selecting-DP stage, or does it undergo further changes?

4) The ChIP-seq and RNA-seq data in Figure 2 have been done on background genotypes (OT-II and OT-II+HDCA3-null) that appear to be different from the ones used in Figure 1. It would be useful to provide data showing convergence between these various background genotypes.

5) In a number of instances, comparisons are offered between OT-II and OT-II-RB3 mice. It would be advisable to include OT-II-RB genotypes in these comparisons, to ensure formally that the major genotypic distinction is indeed in HDAC3.

6) The mechanistic model appears to be HDAC3-IL-21R-STAT5-PIAS. Is PIAS in fact essential for this specific functional role of HDAC3? As a related but separate question, does IL-21R expression and signaling precede the immature-DP super-enhancer alterations? Does the role of IL-21R in regulating genome-wide acetylation patterns indicate that acetylation is not driven directly by HDAC3? And instead, that HDAC3 has a limited role, for this function, in regulating IL-21R expression alone? Evidence for these mechanistic steps in mediating the final functional outcome of lineage choice would be very useful.

7) A cartoon of the mechanistic pathways involved would be very helpful, even if some of the steps in it remain speculative.

*Reviewer #2:*

HDAC3 deficient mice have defects in intrathymic selection, previously studied by these workers. In this work, the authors correct for defects in RORγT down regulation using mice deficient for RORγT and correct for the survival defect resulting from RORγT deficiency by introducing a transgene for Bcl-xl. Such mice were previously shown to produce only CD8 but not CD4 cells. Here, the MHC-II restricted TCR transgenic mice OT-II are shown to develop into the CD8 lineage when RORγT and HDAC3 is missing and a Bcl-xl transgene (OTII-RB3 mice) is present. Subsequent investigation largely uses a simpler OT-II TCR transgenic HDAC3 deficient model, and thymocytes that initiate TCR signaling are isolated and studied. The authors find that HDAC3 deficiency leads to increased H3K27Ac deposition specifically on genes associated with the CD8 lineage. This is an interesting finding. Among these are genes for Runx3, which is critical for commitment to the CD8 lineage. Surprisingly, enhanced Runx3 expression is shown to be secondary to IL-21 signaling, as expression of the IL-21R receptor is enhanced on signaled thymocytes in OTII-RB3 mice, and this enhanced expression is reversed when antibody to IL-21R is injected into OTII-RB3 mice. Antibody blockade of IL-21R also reverses the changes in H3K27Ac, including the increased H3K27Ac at the *Il21r* locus. The authors suggest that HDAC3 is critical to restrain cytokine signaling in thymocytes, and in its absence IL-21 signaling leads to STAT5 activation, and so CD8 lineage commitment. How HDAC3 restrains *Il21r* cytokine signaling is unresolved.

There are observations of interest here. One serious omission is that the authors do not appear to address whether blockage of IL-21R blocks CD8 development in OTII-RB3 mice. Further, the mechanism by which HDAC3 normally suppresses expression of IL-21R is also not resolved, because some measured changes in HDAC deficient cells are shown to be secondary to IL-21R signaling. Experiments would need to be repeated in the absence of IL-21R to address the mechanism of action of HDAC3.

In summary, I believe the work is potentially of interest, but quite preliminary. Further insight into the role of HDAC3 and its putative role in control of IL-21R in thymocyte selection is needed. The importance of IL-21R signaling for CD8 development also needs to be better clarified.

*Reviewer #3:*

Philips et al. explore the role of HDAC3 in CD4/8 lineage commitment and reveal that HDAC3 loss causes a striking redirection of MHC-II restricted TCR Tg DP thymocytes to the CD8 fate. The authors go on to attempt to unravel the molecular basis for this highly penetrant phenotype, revealing changes in the expression of cytokine receptors and in responsiveness to cytokine signaling. Finally, the authors use an antibody blocking strategy to demonstrate the IL-21R signaling is responsible for changes in acetylation and STAT5 binding to the critical regulator of lineage fate, Runx3, in HDAC3-deficient mice. It was then concluded that HDAC3-deficiency regulates lineage commitment through effects on cytokine signaling, mostly likely IL-21R. The experiments are well controlled and thoughtfully interpreted. However, while authors have demonstrated some effects of HDAC3 loss on cytokine signaling, they did not definitively demonstrate that this was the cause of altered lineage fate, as they made no attempt to normalize cytokine signaling in HDAC3 mice and link this to restoration of CD4 lineage commitment. If the effect of HDAC3 on IL-21R expression and responsiveness is thought to play a key role, then it should be possible to extend the IL-21R blockade and examine effects on lineage commitment.

---

## [Author Response]

[Editors’ note: what follows is the authors’ plan to address the revisions.]

Essential revisions:While the manuscript by Philips et al. show data supporting an interesting role for HDAC3 in regulating the genomic preparedness for thymocyte lineage differentiation into the CD4 branch, the data do not yet quite provide for the visualisation of a robust mechanistic pathway, and there are a number of more formal concerns as well. If these limitations are addressed substantively, it would increase enthusiasm for publication in eLife.1) How HDAC3 restrains IL-21R cytokine signaling remains unresolved.

We believe that HDAC3 may directly regulate IL-21R expression, which we will examine. Two groups have published HDAC3 ChIP-seq in either peripheral human CD4 T cells (Wang et al., 2009) or in the murine pro-B cell line Ba/F3 (Nanou et al., 2017). Using the publicly available datasets, we find that in each system, HDAC3 binds to the IL-21R promoter (see Author response image 1). Warren Leonard’s group (Wu et al., 2005) identified that the primary regulator of the IL-21R promoter is Sp1 which binds to the promoter at approximately -45 relative to the transcriptional start site. Several groups have identified that Sp1 and HDAC3 interact, and that HDAC3 interaction with Sp1 inhibits transcriptional activation. To test this, we will:

– perform HDAC3 ChIP-qPCR to determine if HDAC3 interacts directly with the IL-21R promoter, using primers that flank the Sp1 site, and

– determine if HDAC3 overexpression inhibits transcriptional activation of an IL-21R reporter construct.

**Author response image 1. respfig1:** HDAC3 binding at the *Il21r* promoter in human CD4 T cells and mouse pro-B cells. Accession numbers for publicly accessible HDAC3 ChIP-seq: human CD4 T cells (GSM393952) and mouse pro-B cells (GSM2096648).

Also, does IL-21R expression and signaling precede the immature-DP super-enhancer alterations?

To determine whether IL-21R expression and signaling occurs prior to DP thymocytes, we examined IL-21R expression and pSTAT5 signaling in DN (CD4^-^CD8^-^) and immature SP (ISP; CD8^+^CD4^-^TCRb^-^) thymocytes from WT and HDAC3-cKO mice. We found no change in IL-21R expression in DN thymocytes between mice, however ISPs from HDAC3-cKO mice exhibited a significant increase in IL-21R expression compared to WT mice. Of note, the level of IL-21R expression in ISPs from HDAC3-cKO mice was less than what was observed in DP thymocytes. In addition, upon IL-21 stimulation, ISPs from HDAC3-cKO mice exhibited a significant increase in phospho-STAT5 levels compare to WT mice, while stimulation with IL-4, IL-7, and IL-15 showed no effect. Therefore, IL-21R expression and signaling does precede the DP stage. This figure can be added to the supplemental data as Figure 5—figure supplement 2.

Does the role of IL-21R in regulating genome-wide acetylation patterns indicate that acetylation is not driven directly by HDAC3? And instead, that HDAC3 has a limited role, for this function, in regulating IL-21R expression and/or signaling alone?

We have not examined genome-wide acetylation patterns when an IL-21R blocking antibody was used. Select analysis of H3K9Ac at Runx3, Patz1 and IL-21R loci demonstrated decreased histone acetylation when IL-21R blocking antibody was administered, indicating that at least part of the increase in histone acetylation observed in HDAC3-deficient DP thymocytes is due to IL-21R signaling rather than a direct effect of the loss of HDAC3. This can be clarified by altering the Discussion to emphasize this point. For the revision:

– We will examine a larger subset of sites of increased histone acetylation at CD8 lineage genes to determine the relative contribution of IL-21R. The identification of sites of histone acetylation that are independent of IL-21 would demonstrate that there are effects independent of IL-21.

The mechanistic model appears to be HDAC3-IL-21R-STAT5-PIAS. Is PIAS in fact essential for this specific functional role of HDAC3? Evidence for these mechanistic steps in mediating the final functional outcome of lineage choice would be very useful. A cartoon of the mechanistic pathways involved would also be very helpful.

A cartoon of our model is shown in Author response image 2. Our model is that HDAC3 is a direct inhibitor of IL-21R, keeping expression low in WT DP thymocytes. In the absence of HDAC3, IL-21R levels increase, leading to enhanced Stat5 activation, Runx3 expression and CD8 lineage commitment.

We demonstrate there is decreased PIAS3 in HDAC3-deficient DP thymocytes by RNA-seq. In the discussion, we speculate as to a role of PIAS to potentially potentiate Stat5 activation observed due to IL-21R signaling. Direct examination to determine the relative contribution of PIAS3 would require substantial interbreeding beyond the scope of this manuscript and will be addressed in future studies. If desired by the reviewers, the speculation in the discussion can be removed.

**Author response image 2. respfig2:** Model. HDAC3 functions to block expression of IL-21R in DP thymocytes to prevent IL-21R expression, STAT5 activation, and expression of CD8-lineage promoting genes, such as *Runx3*

2) Another serious omission is that the authors do not appear to address whether blockage of IL-21R blocks CD8 development in OTII-RB3 mice. While authors have demonstrated some effects of HDAC3 loss on cytokine signaling, they have not definitively demonstrated that this is the cause of altered lineage fate, as they have made no attempt to normalize cytokine signaling in HDAC3 mice and link this to restoration of CD4 lineage commitment. If the effect of HDAC3 on IL-21R expression and responsiveness is thought to play a key role, then it should be possible to extend the IL-21R blockade and examine effects on lineage commitment.

Blocking antibodies may or may not be sufficient to eradicate all signaling. In order to determine the relative contribution of IL-21 receptor signaling to CD8-lineage commitment in OT-II RB3 mice, we will transfer bone marrow from OT-II RB3 mice (or OT-II controls) into IL-21-cytokine-knockout recipients. Bone marrow chimeras will be analyzed for lineage redirection in the thymus and spleen. A previous study showed that thymic epithelial cells, rather than haematopoietically-derived cells, are the source of IL-21 in the thymus (Rafei et al., 2013). Therefore, using bone marrow chimeras with the host thymic stroma deficient in IL-21 can address whether removing IL-21 may restore CD4-lineage development in OT-II RB3 mice. Cryopreserved IL-21 heterozygous mice were ordered from JAX, and these mice were recently received. We are currently interbreeding in our facility to generate IL-21-homozygous-knockout specifically for this experiment. One caveat is that IL-4R expression is also significantly higher in HDAC3-deficient DP thymocytes (Figure 5A) and IL-4 leads to enhanced Stat5 activation (although this was not quite statistically significant). We will also examine whether there is enhanced IL-4R expression and signaling in the HDAC3-deficient thymocytes in an IL-21-deficient recipient.

3) The functional role of HDAC3 is revealed primarily in RORγT-null Bcl-xl-transgenic mice. While the argument for this complex genotypic background is reasonable, it leaves open an uncertainty about the role of HDAC3 in wild-type double-positive (DP) thymocyte differentiation. Even limited functional data supporting such a role in functional assays in closer-to-wild-type mice would be useful.

There are almost no SP cells generated in CD2-icre HDAC3 cKO mice, due to an inability to pass positive selection (see Author response image 3). Therefore, lineage commitment cannot be examined in this mouse.

**Author response image 3. respfig3:** Very few SP thymocytes are generated in CD2-icre HDAC3-cKO mice. Cell count of CD4SP and CD8SP thymocytes from Rag-GFP and Rag-GFP CD2-icre HDAC3-cKO mice. SP thymocytes were gated on Rag-GFP+ to remove recirculating (GFP-) T cells in the thymus. Graph shows mean +/- SEM of 5 mice/group.

4) The lack of inclusion of a Rag1/2-null genotype in the OT-II experiments (Figure 1) leaves the formal possibility open that endogenous MHC class I-restricted TCRs have provided positive selection into the CD8 lineage. Data to rule out such a possibility would be helpful.

We understand the reviewers’ concerns as we had not interbred to regenerate a Rag-deficient OTII RB3 (CD2-icre Hdac3 cKO RORγT KO Bcl-xl tg). In order to exclude examination of endogenously rearranged TCR in OT-II tg experiments, we utilized a gating scheme that analyzed only Vβ5^+^ Vα2^+^ cells in Figures 1 and 4 to restrict analysis primarily to OTII expressing T cells. Our sorts for ChIP-seq and RNA-seq analysis utilized Vβ5 as shown in Figure 2. Therefore, we do not expect there to be a significant number of endogenously rearranged TCRs that express Vβ5 and Vα2 within these gates. However, if desired by the reviewers, we can perform bone marrow chimeras in which bone marrow from OT-II RB3 mice or OT-II controls will be transferred into irradiated β2m-knockout recipients. Thymocytes from β2m-knockout mice largely fail to positively select MHC class I-restricted TCRs, leading to the development of only CD4 T cells (Broussard et al., 2006, their Figure 4A). We propose the bone marrow chimera experiment, as opposed to introducing a Rag1/2-null genotype into OT-II RB3 mice, as the time it would take to complete the interbreeding to six alleles with additional experiments would not be feasible for this revision.

5) The ChIP-seq and RNA-seq data in Figure 2 have been done on background genotypes (OT-II and OT-II+HDCA3-null) that are different from the ones used in Figure 1. It would be useful to provide data showing convergence between these various background genotypes.

The alterations observed in OT-II HDAC3-cKO mice are also observed in OT-II RB3 mice. The ChIP-seq and RNA-seq demonstrated that prior to positive selection, OT-II HDAC3-cKO mice exhibited a bias for the CD8-lineage. This was also observed at the protein level, where *Selecting* thymocytes (Vβ5^lo^ CD69^+^) from OT-II HDAC3-cKO mice exhibited an increase in Runx3 protein expression and a decrease in ThPOK protein expression compared to OT-II *Selecting* cells (Figure 2—figure supplement 2D). This is consistent with the observation that OT-II RB3 mice exhibit an increase in Runx3 and a decrease in ThPOK expression in CD69-positive CCR7-low thymocytes (Figure 4). In addition, OT-II HDAC3-cKO mice and OT-II RB3 mice both express IL-21R in DP thymocytes at a significantly higher level compare to OT-II mice (see Author response image 4). We will need to perform additional experiments to increase our N for the provided figure (we currently have an N of 2). Furthermore, DP thymocytes from OT-II RB3 mice also show elevated p-STAT5 levels after IL-21 stimulation compared to OT-II mice, similarly to OT-II HDAC3-cKO mice (see Author response image 4). Therefore, OT-II HDAC3-cKO mice and OT-II HDAC3-cKO mice converge in their expression of lineage-determining transcription factors in CD69^+^ cells, increased expression of IL-21R in DP thymocytes, and elevated STAT5 activation in DP cells. This data will be added to the manuscript.

**Author response image 4. respfig4:** OT-II RB3 mice show increased IL-21R expression and IL-21-induced STAT5 activation. (**A**) IL-21R expression on DP thymocytes from OT-I, OT-II, and OT-II RB3 mice. (**B**) p-STAT5 expression in DP cells from OT-I, OT-II, and OT-II RB3 mice that were stimulated with the indicated cytokines for 10mins. Experimental conditions are the same as performed in Figure 5E. Bar graph shows mean +/- SEM. N=2 mice/group.

6) In a number of instances, comparisons are offered between OT-II and OT-II-RB3 mice. It would be advisable to include OT-II-RB genotypes in these comparisons, to ensure formally that the major genotypic distinction is indeed in HDAC3.

In the submitted manuscript, we had included the OT-II RB control when OT-II and OT-II RB3 mice are compared in Figure 1. In the manuscript, the main comparisons made between OT-II and OT-II RB3 mice are (1) CD4-vs-CD8 profile of mature SP and peripheral T cells in Figure 1A, (2) expression of Runx3 and ThPOK in SP thymocytes in Figure 1B, (3) whether SP cells produce granzyme b and perforin in Figure 1D, and (4) the expression of Runx3, CD8α, and ThPOK during the 2 phases of lineage commitment in Figure 4. The manuscript currently includes an OT-II RB control for Figure 1A, however we can add additional controls to the rest of the figures. As shown in Author response image 5, we find that OT-II RB mice downregulate CD8α and fail to induce Runx3 during lineage commitment similarly to OT-II mice. We will need to perform additional experiments to increase our N for measuring CD8α and Runx3 during lineage commitment (for addition to Figure 4, we currently have an N of 2 for OT-II RB). In addition, we will measure granzyme b and perforin production in OT-II RB mice (for addition to the comparison of OT-II and OT-II RB3 in Figure 1D), and ThPOK (to be added to Figure 1B).

**Author response image 5 respfig5:** Acceleration of CD8-lineage commitment in HDAC3-deficient thymocytes. Expression of Runx3 and CD8α during stages of lineage commitment (Stages 1-6), from OT-I, OT-II, and OT-II RB3 straight bone marrow chimeras. CD69-versus-CCR7 plots were gated from DN-removed, CD45.2^+^ cells. Plots show mean ± SEM of MFI from 4-5 mice per group from three independent experiments, except N=2 mice for OT-II RB.

[Editors’ note: what now follows is the decision letter after the authors submitted for further consideration.]

Essential revisions:While the manuscript by Philips et al. show data supporting an interesting role for HDAC3 in regulating the genomic preparedness for thymocyte lineage differentiation into the CD4 branch, the data do not yet quite provide for the visualisation of a robust mechanistic pathway, and there are a number of more formal concerns as well. If these limitations are addressed substantively, it would increase enthusiasm for publication in eLife.

As the revision of our original submission required bone marrow chimera experiments to address a reviewer’s concern, as well as additional breeding of mice with complex genotypes, we could not complete our revision in the 2-month time limit for *eLife*. However, we welcomed the opportunity to revise and submit our manuscript after this window. We appreciate the time and effort devoted by the reviewers and editors.

As γc cytokines have an established and critical role in CD8 lineage commitment, we initially focused on the role of enhanced IL-21R expression in CD8 lineage commitment. Based on the reviews, a longer 14-day time course of IL-21R blockade was performed to analyze changes in CD4/CD8 lineage commitment, ThPOK and Runx3 expression in OT-II and OT-II RB3 mice. 14-day blockade of IL-21R did not alter OT-II RB3 lineage commitment to CD8 or modulate expression of either ThPOK or Runx3. This data has been added as Figure 6, and the previous data on changes in histone acetylation has been removed. Therefore, blockade of IL-21R is insufficient to normalize lineage commitment in the absence of HDAC3, and HDAC3 must have additional mechanisms for regulation of CD8 lineage commitment. We have now added HDAC3 ChIP-qPCR studies, and demonstrate that HDAC3 binds to the Runx3, Patz1 and IL-21R loci in DP thymocytes, showing a direct role for HDAC3 in the regulation of these genes. In addition, we examined the binding of HDAC3 at Runx3 and Patz1 loci in (HDAC3-sufficient) OT-I CD8 SP thymocytes and OT-II CD4-SP thymocytes. Similar to WT DP thymocytes, HDAC3 remained bound to Runx3 and Patz1 loci in OT-II CD4 SP thymocytes. However, very little HDAC3 bound to the Runx3 and Patz1 loci in OT-I CD8 SP thymocytes, indicating that HDAC3 must be removed from these loci in the CD8 lineage. This data has been added as Figure 7 in the manuscript. Therefore, we conclude that HDAC3 restrains multiple CD8-lineage genes in DP thymocytes to allow for appropriate CD4/CD8 lineage commitment.

What follows is our point-by-point response to the Essential Revisions.

1) How HDAC3 restrains IL-21R cytokine signaling remains unresolved.

We now show that HDAC3 directly associates with the IL-21R promoter by qChIP.

Also, does IL-21R expression and signalling precede the immature-DP super-enhancer alterations?

To determine whether IL-21R expression and signaling occurs prior to DP thymocytes, we examined IL-21R expression and pSTAT5 signaling in DN (CD4^-^CD8^-^) and immature SP (ISP; CD8^+^CD4TCRb^-^) thymocytes from WT and HDAC3-cKO mice. We found no change in IL-21R expression in DN thymocytes between mice, however ISPs from HDAC3-cKO mice exhibited a significant increase in IL-21R expression compared to WT mice (now added as new Figure 5—figure supplement 2). Of note, the level of IL-21R expression in ISPs from HDAC3-cKO mice was less than what was observed in DP thymocytes. In addition, upon IL-21 stimulation, ISPs from HDAC3cKO mice exhibited a significant increase in phospho-STAT5 levels compare to WT mice, while stimulation with IL-4, IL-7, and IL-15 showed no effect (included in new Figure 5—figure supplement 2). Therefore, IL-21R expression and signaling does precede the DP stage.

Does the role of IL-21R in regulating genome-wide acetylation patterns indicate that acetylation is not driven directly by HDAC3? And instead, that HDAC3 has a limited role, for this function, in regulating IL-21R expression and/or signaling alone?

14-day IL-21R blockade did not revert lineage commitment or altered ThPOK or Runx3 expression in OT-II RB3 mice. Thus, the function of HDAC3 in lineage commitment is not solely due to upregulation of IL-21R. As mentioned above, we now demonstrate that HDAC3 directly binds to Runx3 and Patz1 loci by ChIP. Therefore, HDAC3 constrains multiple CD8 lineage genes in DP thymocytes and that HDAC3 is required for DP thymocytes to maintain a bi-potential state that allows subsequent CD4 lineage commitment for Class II-restricted, positively-selected thymocytes.

The mechanistic model appears to be HDAC3-IL-21R-STAT5-PIAS. Is PIAS in fact essential for this specific functional role of HDAC3? Evidence for these mechanistic steps in mediating the final functional outcome of lineage choice would be very useful.

We have removed speculation on the role of PIAS from the manuscript.

A cartoon of the mechanistic pathways involved would also be very helpful.

A model has been added as new Figure 7—figure supplement 2.

2) Another serious omission is that the authors do not appear to address whether blockage of IL-21R blocks CD8 development in OTII-RB3 mice. While authors have demonstrated some effects of HDAC3 loss on cytokine signaling, they have not definitively demonstrated that this is the cause of altered lineage fate, as they have made no attempt to normalize cytokine signaling in HDAC3 mice and link this to restoration of CD4 lineage commitment. If the effect of HDAC3 on IL-21R expression and responsiveness is thought to play a key role, then it should be possible to extend the IL-21R blockade and examine effects on lineage commitment.

As stated above, 14-day IL-21R blockade did not restore lineage commitment in OTII-RB3 mice. IL-21R blockade also did not normalize expression of Runx3, as Runx3 protein expression was still increased in the absence of HDAC3. We now show by HDAC3 ChIP-qPCR that HDAC3 binds to Runx3, Patz1 and IL-21R loci. Therefore, there multiple effects of HDAC3 genes that regulate CD8 lineage commitment, and blocking IL-21R alone is insufficient. The IL-21R blockade data has been added to Figure 6 and HDAC3 qChIP added to Figure 7.

3) The functional role of HDAC3 is revealed primarily in RORγT-null Bcl-xl-transgenic mice. While the argument for this complex genotypic background is reasonable, it leaves open an uncertainty about the role of HDAC3 in wild-type double-positive (DP) thymocyte differentiation. Even limited functional data supporting such a role in functional assays in closer-to-wild-type mice would be useful.

There are almost no SP cells generated in CD2-icre HDAC3 cKO mice, due to an inability to pass positive selection (see Author response image 3). Therefore, lineage commitment cannot be examined in this mouse.

However, to understand the role of HDAC3 in WT DP thymocyte differentiation, we performed HDAC3 qChIP at the Runx3 and Patz1 loci in WT DP thymocytes (Figure 7A), OT-II CD4 SP thymocytes (Figure 7D) and OT-I CD8 SP thymocytes. HDAC3 binds to the Runx3 and Patz1 loci in WT DP thymocytes, and this binding is maintained in OT-II CD4 SP thymocytes. However, there is a dramatic decrease in HDAC3 binding to the Runx3 and Patz1 loci in OT-I CD8 SP thymocytes, indicating that HDAC3 is removed from these sites in the normal process of CD8 lineage commitment but maintained in CD4 lineage commitment.

4) The lack of inclusion of a Rag1/2-null genotype in the OT-II experiments (Figure 1) leaves the formal possibility open that endogenous MHC class I-restricted TCRs have provided positive selection into the CD8 lineage. Data to rule out such a possibility would be helpful.

We understand the reviewers’ concerns as we had not interbred to regenerate a Rag-deficient OTII RB3 (CD2-icre Hdac3 cKO RORγt KO Bcl-xl tg). In order to exclude examination of endogenously rearranged TCR in OT-II tg experiments, we utilized a gating scheme that analyzed only Vβ5^+^ Vα2^+^ cells in Figures 1 and 4 to restrict analysis primarily to OTII expressing T cells. Our sorts for ChIP-seq and RNA-seq analysis utilized Vβ5 as shown in Figure 2. Therefore, we did not expect there to be a significant number of endogenously rearranged TCRs that express Vβ5 and Vα2 within these gates. To address this point, we generated bone marrow chimeras in which bone marrow from OT-II RB3 mice or OT-II controls were transferred into irradiated β2m-knockout recipients. Thymocytes from β2m-knockout mice largely fail to positively select MHC class I-restricted TCRs, leading to the development of only CD4 T cells (Broussard et al., 2006, their Figure 4A).

OTII-RB3 thymocytes within a β2m-knockout environment generated CD8^+^ mature SP thymocytes, similarly to OTII-RB3 thymocytes in a WT environment. Thus, the effects that we observe are not due to endogenous MHC class-I restricted TCRs generated CD8 SP thymocytes in OTII-RB3 mice. These results have been added in Figure 1E in the manuscript.

5) The ChIP-seq and RNA-seq data in Figure 2 have been done on background genotypes (OT-II and OT-II+HDCA3-null) that are different from the ones used in Figure 1. It would be useful to provide data showing convergence between these various background genotypes.

The alterations observed in OT-II HDAC3-cKO mice are also observed in OT-II RB3 mice. The ChIP-seq and RNA-seq demonstrated that prior to positive selection, OT-II HDAC3-cKO mice exhibited a bias for the CD8-lineage. This was also observed at the protein level, where *Selecting* thymocytes (Vβ5^lo^ CD69^+^) from OT-II HDAC3-cKO mice exhibited an increase in Runx3 protein expression and a decrease in ThPOK protein expression compared to OT-II *Selecting* cells (Figure 3—figure supplement 2D). This is consistent with the observation that OT-II RB3 mice exhibit an increase in Runx3 and a decrease in ThPOK expression in CD69-positive CCR7-low thymocytes (Figure 2B). In addition, HDAC3-cKO mice and OT-II RB3 mice both express IL-21R in DP thymocytes at a significantly higher level compare to OT-II mice (Figure 5). Furthermore, DP thymocytes from OT-II RB3 mice also show elevated p-STAT5 levels after IL-21 stimulation compared to OT-II mice, similarly to HDAC3-cKO mice (now added as Figure 5—figure supplement 1). Therefore, OT-II HDAC3-cKO mice and OT-II RB3 mice converge in their expression of lineage-determining transcription factors Runx3 and ThPOK in CD69^+^ cells, increased expression of IL-21R in DP thymocytes, and elevated STAT5 activation in DP cells.

6) In a number of instances, comparisons are offered between OT-II and OT-II-RB3 mice. It would be advisable to include OT-II-RB genotypes in these comparisons, to ensure formally that the major genotypic distinction is indeed in HDAC3.

We have added data as requested on OTII-RB mice. In the current manuscript, OT-II, OTII-RB and OT-II RB3 mice are now compared side-by-side in (1) CD4-vs.-CD8 profile of mature SP and peripheral T cells in Figure 1A, (2) expression of Runx3 and ThPOK in SP thymocytes in Figure 1B, (3) granzyme b and perforin expression after stimulation in Figure 1D, (4) the expression of Runx3,

CD8α, and ThPOK during the kinetics of lineage commitment in Figure 2 and (5) examination of IL-21R and p-STAT5 activation in Figure 5—figure supplement 1. In all cases, OT-II RB mice are similar to OT-II mice with respect to appropriate CD4 lineage commitment, demonstrating that the altered lineage commitment to CD8 in OT-II RB3 mice is due to loss of HDAC3.

References:

Broussard C, Fleischacker C, Horai R, Chetana M, Venegas AM, Sharp LL, Hedrick SM, Fowlkes BJ, Schwartzberg PL. Altered development of CD8^+^ T cell lineages in mice deficient for the Tec kinases Itk and Rlk. Immunity (2006) 25: 93-104. DOI: 10.1016/j.immuni.2006.05.011

Rafei M, Rouette A, Brochu S, Vanegas JR, Perreault C. Differential effects of γc cytokines on postselection differentiation of CD8 thymocytes, Blood (2013) 121:107-117. DOI: 10.1182/blood-2012-05-433508

Wu Z1, Kim HP, Xue HH, Liu H, Zhao K, Leonard WJ. Interleukin-21 receptor gene induction in human T cells is mediated by T-cell receptor-induced Sp1 activity Mol. Cell Biol. (2005) 25:9741-9752. DOI: 10.1128/MCB.25.22.9741-9752.2005